# Selective Explanations

**Lucas Monteiro Paes**
Harvard University
lucaspaes@g.harvard.edu

**Dennis Wei**
IBM Research
dwei@us.ibm.com

**Flavio P. Calmon**
Harvard University
flavio@seas.harvard.edu

## Abstract

Feature attribution methods explain black-box machine learning (ML) models by assigning importance scores to input features. These methods can be computationally expensive for large ML models. To address this challenge, there have been increasing efforts to develop *amortized explainers*, where a ML model is trained to efficiently approximate computationally expensive feature attribution scores. Despite their efficiency, amortized explainers can produce misleading explanations. In this paper, we propose *selective explanations* to (i) detect when amortized explainers generate inaccurate explanations and (ii) improve the approximation of the explanation using a technique we call *explanations with initial guess*. Selective explanations allow practitioners to specify the fraction of samples that receive explanations with initial guess, offering a principled way to bridge the gap between amortized explainers (one inference) and more computationally costly approximations (multiple inferences). Our experiments on various models and datasets demonstrate that feature attributions via selective explanations strike a favorable balance between explanation quality and computational efficiency.

## 1 Introduction

Large black-box models are increasingly used to support decisions in applications ranging from online content moderation [26], hiring [12], and medical diagnostics [35]. In such high-stakes settings, the need to explain "why" a model produces a given output has led to a growing number of perturbation-based *feature attribution* methods [22, 29, 27, 23, 2, 40]. These methods use input perturbations to assign numerical values to each input feature (e.g., words in a text) a model uses, indicating their influence on the model prediction. They are widely adopted in part because they work in the black-box setting with access only to model output (i.e., without gradients). However, existing feature attribution methods can be prohibitively expensive for the large models used in the current machine learning landscape (e.g., language models with billions of parameters) since they require a significant number of inferences for each individual explanation.

Recent literature has introduced two main *approximation* strategies to speed up existing feature attribution methods for large models: (i) employing Monte Carlo methods to approximate explanations with fewer computations [22, 29, 5, 24], and (ii) adopting an *amortized* approach, training a separate model to "mimic" the outputs of a reference explanation method [16, 6, 38, 32, 3, 33]. Monte Carlo approximations can yield accurate approximations for attributions but may converge slowly, limiting their practicality for and online applications. In contrast, amortized explainers require only one inference per explanation, making them efficient for large black-box models and online explanations. However, as shown in Figure 1, amortized explainers can produce highly diverging explanations from their reference due to lack of precision in approximations. Aiming to benefit from Monte Carlo and amortized explainers, we propose *selective explanations* to answer the questions:

**(Q1)** When are amortized explanations inaccurate ?

**(Q2)** How can we improve inaccurate amortized explanations using additional computations?

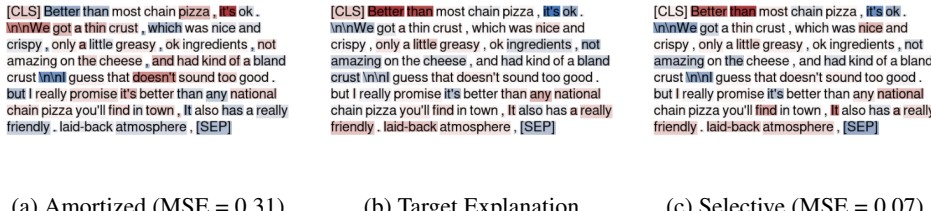

(a) Amortized (MSE = 0.31)   (b) Target Explanation   (c) Selective (MSE = 0.07)

Fig. 1: Amortized explainer (a) compared with a target explainer (SHAP [22]) (b) and our selective explanation method (c). All methods flag input parts that contribute to the `YelpLLM` predicting the given example is a `Negative Review`. We observe that both target and selective explanations attribute "not amazing" for the negative review (blue), while the amortized explainer misses this term.

To answer (Q1) and (Q2), we propose *selective explanations*, a method that bridges Monte Carlo and amortized explanations. The selective explainer first trains a model that "learns to select" which data points will receive inaccurate amortized explanations, and then performs additional computations to further approximate target explanations. The key idea behind the selective explanation method is to use Monte Carlo explanations only for points that would receive inaccurate amortized explanations; see Figure 2 for the workflow of selective explanations. The code for generating selective explanations can be found at https://github.com/LucasMonteiroPaes/selective-explanations.

The ideas of predicting selectively and providing recourse with a more accurate but expensive method (usually human feedback) have been explored in classification and regression [28, 10, 7, 9, 11]. To our knowledge, however, these ideas have not been applied to feature attribution methods. We make **two contributions** in this regard that are relevant for selective prediction more generally. (1) Selective prediction uses *quality metrics* to identify input points for which the predictor (the amortized explainer in our case) would produce inaccurate outputs and recourse is needed. The high-dimensional nature of explanations requires us to develop new quality metrics (Section 3) suitable for this setting. (2) Instead of providing recourse with a Monte Carlo explanation alone, we use an optimized method called *explanations with initial guess* (Section 4) that combines amortized and Monte Carlo explanations in a optimized manner, improving the approximation to the target explanation beyond that of either method individually.

Our **overall contribution** (3) is to combine (1) and (2) in the form of *selective explanations*, providing explanations with initial guess to improve inaccurate amortized explanations. We validate our selective explanations approach on two language models as well as tabular datasets demonstrating its ability to (i) detect inaccurate explanations from the amortized explainer, (ii) enhancing amortized explanations even when Monte Carlo explanations are inaccurate, and (iii) improving the worst explanations from the amortized model.

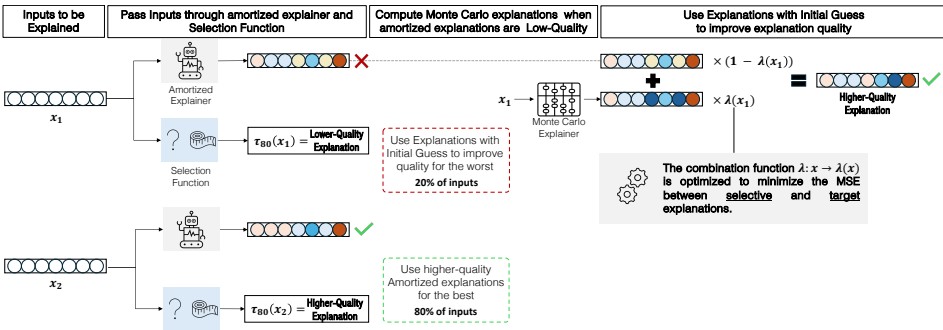

Fig. 2: Workflow of selective explanations.

## 2 Problem Setup & Background

We aim to explain the predictions of a fixed probabilistic black-box model $h$ that predicts $h(\boldsymbol{x}) = (h_1(\boldsymbol{x}), ..., h_{|\mathcal{Y}|}(\boldsymbol{x}))$ and outputs $\operatorname{argmax}_{j \in \mathcal{Y}} h_j(\boldsymbol{x}) \in \mathcal{Y}$ using a vector of features $\boldsymbol{x} = (x_1, ..., x_d) \in \mathbb{R}^d$. The user specifies an output of interest $\boldsymbol{y} \in \mathcal{Y}$ (usually $\boldsymbol{y} = \operatorname{argmax}_{j \in \mathcal{Y}} h_j(\boldsymbol{x})$) and our goal is to efficiently explain *Why would $h$ output $\boldsymbol{y}$ for a given $\boldsymbol{x}$?* We consider a dataset $\mathcal{D} = \{(\boldsymbol{x}_i, \boldsymbol{y}_i)\}_{i=1}^N$ comprised of $N > 0$ samples divided into three parts: $\mathcal{D}_{\texttt{train}}$ for training $h$ and the explainers, $\mathcal{D}_{\texttt{cal}}$ for calibration and validation, and $\mathcal{D}_{\texttt{test}}$ for testing. Thus, $\mathcal{D} = \mathcal{D}_{\texttt{train}} \cup \mathcal{D}_{\texttt{cal}} \cup \mathcal{D}_{\texttt{test}}$. Moreover, for a subset $S = \{i_1, ..., i_{|S|}\} \subset [d]$ we write $\boldsymbol{x}_S \triangleq (x_{i_1}, ..., x_{i_{|S|}})$.

**Feature Attribution Methods,** also called *explainers*, are functions $\mathbb{R}^d \times \mathcal{Y} \to \mathbb{R}^d$ that assess the importance of each feature for the model's ($h$) prediction to be $\boldsymbol{y}$ for a given input vector $\boldsymbol{x}$. We consider three types of explainers:

   (i) **Target explainers** that use a large number of computations to provide explanations (e.g., SHAP with $2^d$ inferences from model $h$) [22, 29], denoted by $\mathsf{Target}(\boldsymbol{x}, \boldsymbol{y})$;

  (ii) **Monte Carlo explainers** that approximate fixed target explainers using $n$ inferences from model $h$ per explanation [22, 24], denoted by $\mathsf{MC}^n(\boldsymbol{x}, \boldsymbol{y})$;

 (iii) **Amortized explainers** are trained to approximate the target explanations using only one inference [6, 38], denoted by $\mathsf{Amor}(\boldsymbol{x}, \boldsymbol{y})$.

**Remark 1.** Monte Carlo and amortized explainers aim to approximate the target explanation and are benchmarked on this approximation. We evaluate the performance of Monte Carlo and amortized explainers by computing their distance and correlation to $\mathsf{Target}(\boldsymbol{x}, \boldsymbol{y})$. The usefulness of target explanations (e.g.: SHAP and Lime) has been validated by user studies and automated metrics in [22, 29, 13, 37, 30, 31]. Therefore, we call **higher-quality** the explanations that closely approximate the computationally expensive target and **lower-quality** the one that diverge from the target.

In practice, we measure the quality of a given explanation that aims to approximate the target explanation using a loss (or distortion) function $\ell : \mathbb{R}^d \times \mathbb{R}^d \to \mathbb{R}$, e.g., mean square error (MSE) and Spearman's correlation. The goal of selective and Monte Carlo explanations is to approximate the target explanations while decreasing the number of computations, i.e., to minimize $\ell(\mathsf{SE}(\boldsymbol{x}, \boldsymbol{y}), \mathsf{Target}(\boldsymbol{x}, \boldsymbol{y}))$ with few model inferences.

We define *selective explainers* to provide better approximations to target explanations bridging the gap between Monte Carlo and amortized explainers.

**Definition 1** (**Selective Explainer**). For a given model $h$, an amortized explainer $\mathsf{Amor}$, a Monte Carlo explainer $\mathsf{MC}^n$, a *combination function* $\lambda_h : \mathbb{R}^d \to \mathbb{R}$, and a *selection function* $\tau_\alpha : \mathbb{R}^d \to \{0, 1\}$ (parametrized by $\alpha$), we define the *selective explainer* $\mathsf{SE}(\boldsymbol{x}, \boldsymbol{y})$ as

$$\mathsf{SE}(\boldsymbol{x}, \boldsymbol{y}) \triangleq \begin{cases} \mathsf{Amor}(\boldsymbol{x}, \boldsymbol{y}) & \text{, if } \tau_\alpha(\boldsymbol{x}) = 1, \\ \lambda_h(\boldsymbol{x})\mathsf{Amor}(\boldsymbol{x}, \boldsymbol{y}) + (1 - \lambda_h(\boldsymbol{x}))\mathsf{MC}^n(\boldsymbol{x}, \boldsymbol{y}) & \text{, if } \tau_\alpha(\boldsymbol{x}) = 0. \end{cases} \tag{1}$$

When $\tau_\alpha = 0$, selective explanations output *explanations with initial guess* (Definition 2). Explanations with initial guess linearly combine amortized and Monte Carlo explanations to leverage information from both and provide higher-quality explanations than either explainer alone. Selective explanations heavily depend on three objects that we define in this work and that are covered in the rest of the paper: (i) an uncertainty metric (Section 3), (ii) a selection function (Section 3), and (iii) a combination function (Section 4).

- **Uncertainty metrics** ($s_h$) output the likelihood of the amortized explainer producing a low-quality explanation for an input. Lower $s_h(\boldsymbol{x})$ indicates a higher-quality explanation for $\boldsymbol{x}$. We propose two uncertainty metrics: Deep and Learned Uncertainty (Section 3).

- **Selection function** ($\tau_\alpha$) is a binary rule that outputs 1 for higher-quality amortized explanations and 0 for lower-quality ones based on the uncertainty metric. We define $\tau_\alpha$ to ensure a fraction $\alpha$ of inputs receive amortized explanations. Smaller $\alpha$ implies higher-quality selective explanations but also more computations (Section 3).

- **Combination function** ($\lambda_h$) optimally linearly combines amortized and Monte Carlo explanations to minimize MSE from target explanations (Theorem 1). We propose explanations with initial guess and fit $\lambda_h$ to optimize explanation quality (Section 4).

---

**Algorithm 1** Building a Selective Explainer

---

**Require:** Datasets: $\mathcal{D}_{\texttt{train}}, \mathcal{D}_{\texttt{cal}}$. Explainers: $\mathsf{Amor}, \mathsf{MC}^n, \mathsf{MC}^{n'}$. Coverage: $\alpha$.
**Ensure:** Selection function: $\tau_\alpha$. Combination function: $\lambda_h$.
 1: Fit the uncertainty metric $s_h$ using $\mathcal{D}_{\texttt{train}}$, $\mathsf{Amor}$, and $\mathsf{MC}^n$ (using (4) or (5))
 2: Compute $t_\alpha$ using $\mathcal{D}_{\texttt{cal}}$ (7)
 3: Define the selection function $\tau_\alpha$ using $s_h$ and $t_\alpha$ (6)
 4: Define bins $Q_i = [t_{\alpha_i}, t_{\alpha_{i+1}})$ for partition $\alpha_i = \frac{i-1}{k}$ for $i \in [k+1]$ (9)
 5: For $i \in [k+1]$ Compute $\lambda_i$ as in (12) using $\mathcal{D}_{\texttt{cal}}$, $\mathsf{Amor}$, $\mathsf{MC}^n$, and $\mathsf{MC}^{n'}$.
 6: Define $\lambda_h(\boldsymbol{x}) = \sum_{i=1}^{k+1} \lambda_i \mathbf{1}[s_h(\boldsymbol{x}) \in Q_i]$ as in (9)
 7: **return** $\tau_\alpha, \lambda_h(\boldsymbol{x})$

---

Algorithm 1 describes the procedure to compute the uncertainty metric, selection function, and combination function using the results we describe in Section 3 and 4. Although selective explanations can be applied to any feature attribution method, we focus on Shapley values since they are widely used and most amortized explainers are tailored for them [16, 38, 6]. We discuss how selective explanations can be applied to LIME and provide more details on feature attribution methods in Appendix B. Next, we describe specific feature attribution methods that we use as building blocks for selective explainers of the form (1).

**Shapley Values (SHAP)** [22] is a **target** explainer that attributes a value $\phi_i$ for each feature $x_i$ in $\boldsymbol{x} = (x_1, ..., x_d)$ which is the marginal contribution of feature $x_i$ if the model was to predict $\boldsymbol{y}$

$$\phi_i(\boldsymbol{x}, \boldsymbol{y}) = \frac{1}{d} \sum_{S \subset [d]/\{i\}} \binom{d-1}{|S|}^{-1} \left( h_{\boldsymbol{y}}(\boldsymbol{x}_{S \cup \{i\}}) - h_{\boldsymbol{y}}(\boldsymbol{x}_S) \right). \tag{2}$$

SHAP has several desirable properties and is widely used. However, as (2) indicates, computing Shapley values and the attribution vector $\mathsf{Target}(\boldsymbol{x}, \boldsymbol{y}) = (\phi_1(\boldsymbol{x}, \boldsymbol{y}), ..., \phi_d(\boldsymbol{x}, \boldsymbol{y}))$ requires $2^d$ inferences from $h$, making SHAP impractical for large models where inference is costly. This has motivated several approximation methods for SHAP, discussed next.

**Shapley Value Sampling (SVS)** [24] is a **Monte Carlo** explainer that approximates SHAP by restricting the sum in (2) to $m$ uniformly sampled permutations of features performing $n = md + 1$ inferences. We denote SVS that samples $m$ feature permutations as SVS-$m$.

**Kernel Shap (KS)** [22] is a **Monte Carlo** explainer that approximates Shapley values using the fact that SHAP can be computed by solving a weighted linear regression problem using $n$ input perturbations resulting in $n$ inferences. We refer to Kernel Shap using $n$ inferences as KS-$n$.

**Stochastic Amortization** [6] is an **amortized** explainer that uses noisy Monte Carlo explanations to learn target explanations. Covert et al. [6] trained an amortized explainer in a model class $\mathcal{F}$ (multilayer perceptrons) $\mathsf{Amor} \in \mathcal{F}$ to take $(\boldsymbol{x}, \boldsymbol{y})$ and predicts an explanation $\mathsf{Amor}(\boldsymbol{x}, \boldsymbol{y}) \approx \mathsf{Target}(\boldsymbol{x}, \boldsymbol{y})$ by minimizing the $L_2$ norm from Monte Carlo explanations $\mathsf{MC}^n(\boldsymbol{x}, \boldsymbol{y})$. Specifically, the amortized explainer is given by

$$\mathsf{Amor} \in \operatorname*{argmin}_{f \in \mathcal{F}} \sum_{(\boldsymbol{x}, \boldsymbol{y}) \in \mathcal{D}_{\text{train}}} \| f(\boldsymbol{x}, \boldsymbol{y}) - \mathsf{MC}^n(\boldsymbol{x}, \boldsymbol{y}) \|_2^2. \tag{3}$$

**Amortized Shap for LLMs** [38] is a **amortized** explainer similar to stochastic amortization but tailored for LLMs. Yang et al. [38] train a linear regression on the LLM embeddings $[e_1(\boldsymbol{x}), ..., e_{|\boldsymbol{x}|}(\boldsymbol{x})]$ to minimize the $L_2$ norm from Monte Carlo explanations $\mathsf{MC}^n(\boldsymbol{x}, \boldsymbol{y})$ and define the amortized explainer as $\mathsf{Amor}(\boldsymbol{x}, \boldsymbol{y}) = (W_{\boldsymbol{y}} e_1(\boldsymbol{x}) + b_{\boldsymbol{y}}, ..., W_{\boldsymbol{y}} e_{|\boldsymbol{x}|}(\boldsymbol{x}) + b_{\boldsymbol{y}})$, where $W_{\boldsymbol{y}}$ is a matrix and $b_{\boldsymbol{y}} \in \mathbb{R}$.

We use stochastic amortization to produce amortized explainers for tabular datasets and amortized Shap for LLMs to produce explainers for LLM predictions. Both explainers are trained using SVS-12 as $\mathsf{MC}^n$. High-quality and Monte Carlo explanations are computed using the Captum library [18].

# 3   Selecting Explanations

This section defines key concepts for selective explainers: (i) uncertainty metrics $s_h$ for amortized explanations and (ii) selection functions ($\tau_\alpha$) to predict when amortized explanations closely approximate target explanations based on the uncertainty metrics.

**Uncertainty Metrics for High-Dimensional Regression:**   An uncertainty metric is a function tailored for the model $h$ that takes $\boldsymbol{x}$ and outputs a real number $s_h(\boldsymbol{x})$ that encodes information about the uncertainty of the model $h$ in the prediction for $\boldsymbol{x}$. Generally, if $s_h(\boldsymbol{x}) < s_h(\boldsymbol{x}')$ then the model is more confident about the prediction $h(\boldsymbol{x})$ than $h(\boldsymbol{x}')$ [10, 28]. Existing uncertainty metrics cater to (i) classification [28, 10, 7, 9, 11] and (ii) one-dimensional regression [39, 34, 11, 17], but none specifically address high-dimensional regression – which is our case of interest ($d$-dimensional explanations). Next, we propose two uncertainty metrics tailored to high-dimensional outputs: (i) Deep uncertainty and (ii) Learned uncertainty.

**Deep Uncertainty** is inspired by deep ensembles [19], a method that uses an ensemble of models to provide confidence intervals for the predictions of one model. We run the training pipeline for the amortized explainer described in (3) $k$ times, each with a different random seed, resulting in $k$ different amortized explainers $\mathsf{Amor}^1, ..., \mathsf{Amor}^k$. We define the deep uncertainty as

$$s_h^{\mathtt{Deep}}(\boldsymbol{x}) \triangleq \frac{1}{dk} \sum_{i=1}^{d} \mathsf{Var}\left(\mathsf{Amor}^1(\boldsymbol{x})_i, ..., \mathsf{Amor}^k(\boldsymbol{x})_i\right). \tag{4}$$

Here, $\mathsf{Var}\left(a_1, ..., a_k\right)$ is the variance of the sample $\{a_1, ..., a_k\}$ and $\mathsf{Amor}^j(\boldsymbol{x})_i$ indicates the $i$-th entry of the feature attribution vector $\mathsf{Amor}^j(\boldsymbol{x})$. Hence, deep uncertainty is the average (across entries) of the variance (across all trained amortized explainers) for the predicted attributions.

If the deep uncertainty for a point $\boldsymbol{x}$ is zero, then the amortized explainers produce the same feature attribution. On the other hand, if the deep uncertainty is high, then the feature attributions vary widely across the amortized explainers. Intuitively, the points with a higher deep uncertainty are more affected by a random seed change, implying more uncertainty in the explanation.

While the Deep Uncertainty approach offers a principled method for estimating the uncertainty of the amortized explainer by leveraging an ensemble of $k$ models, it is computationally expensive due to the need for training, serving, and running multiple models. This overhead can be prohibitive in practice, especially for large-scale applications. To mitigate this issue, we propose *Learned Uncertainty*, which, although less grounded, requires training and serving only a single model.

**Learned Uncertainty** uses data to predict the amortized explainer uncertainty at an input point $\boldsymbol{x}$. We choose $\ell$ (the loss function) between two explanations to be MSE. The learned uncertainty metric is a function in the class $\mathcal{F}$ (multilayer perceptron in our experiments) such that

$$s_h^{\mathtt{Learn}} \in \underset{s \in \mathcal{F}}{\arg\min} \sum_{(\boldsymbol{x}, \boldsymbol{y}) \in \mathcal{D}_{\mathrm{train}}} |s(\boldsymbol{x}) - \ell\left(\mathsf{Amor}(\boldsymbol{x}; \boldsymbol{y}), \mathsf{MC}^n(\boldsymbol{x}; \boldsymbol{y})\right)|^2. \tag{5}$$

Ideally, instead of using the Monte Carlo explanation $\mathsf{MC}^n$ as the reference in (5), we would like to use target explanations, i.e., $\ell\left(\mathsf{Amor}(\boldsymbol{x}; \boldsymbol{y}), \mathsf{Target}(\boldsymbol{x}; \boldsymbol{y})\right)$. However, these computationally expensive explanations are usually not available. Thus, we resort to using Monte Carlo explanations.

For large language models, the textual input $\boldsymbol{x}$ is encoded in a sequence of token embedding $[e_1(\boldsymbol{x}), ..., e_{|\boldsymbol{x}|}(\boldsymbol{x})]$ such that $e_i(\boldsymbol{x}) \in \mathbb{R}^d$ for $i \in [|\boldsymbol{x}|]$. In this case, we use the mean (i.e., "mean-pooling") of the token embeddings to train the learned uncertainty metric instead of $\boldsymbol{x}$.

We analyze the performance of the proposed uncertainty metrics in Section 5.1, showing that it can be used to detect inaccurate explanations from the amortized explainer. Our results indicate that the proposed uncertainty metrics are (i) strongly correlated with how accurate amortized explanations are and (ii) closely approximate the best possible uncertainty measure – the Oracle with knowledge of the approximation quality (Figure 3). Next, we define the selection function that allows practitioners to set a coverage (percentage of points) $\alpha$ that will receive amortized explanations.

**Selection functions:** a selection function is the binary qualifier $(\tau_\alpha)$ that thresholds the uncertainty metric by $t_\alpha \in \mathbb{R}$ given by

$$\tau_\alpha(\boldsymbol{x}) \triangleq \begin{cases} 1 & \text{if } s_h(\boldsymbol{x}) \leq t_\alpha \text{ (high-quality approximations)} \\ 0 & \text{if } s_h(\boldsymbol{x}) > t_\alpha \text{ (low-quality approximations)} \end{cases}. \tag{6}$$

Intuitively, $t_\alpha$ is the maximum uncertainty level tolerated by the user. In practice, if the output of the selection function is 1 (high-quality approximations), we use the explanations from the amortized model because it is probably close to the target; if the output of the selection function is 0 (low-quality approximations), we use explanations with initial guess (see Definition 2 bellow) to improve the explanation provided to the user. The threshold $t_\alpha$ is chosen to be the $\alpha$-quantile of the uncertainty metric to ensure that at least a fraction $\alpha$ of points receive a computationally cheap explanation – $\alpha$ is the *coverage*. Specifically, given $\alpha$, we calibrate $t_\alpha$ in the calibration dataset $\mathcal{D}_{\texttt{cal}}$ and compute it as

$$t_\alpha \triangleq \min_{t \in \mathbb{R}} t, \text{ such that } \Pr_{\texttt{cal}}[s_h(\boldsymbol{x}) \leq t] \geq \alpha, \tag{7}$$

where $\Pr_{\texttt{cal}}$ is the empirical distribution of the calibration dataset. For discussions on selecting coverage with guarantees on the number of inferences for selective explanations, see Appendix C.

**Remark 2.** A property of selective predictions [10], which is transferred to selective explanations, is that it is possible to control the explainer's performance via the threshold $t_\alpha$ with guaranteed performance without providing predictions for all points. This result is displayed in Figure 3.

## 4 Explanations with Initial Guess

We have introduced methods to detect points likely to receive amortized explanations that poorly approximate the target. This raises the question: *How can we improve the explanations for these points?* One approach is to simply use Monte Carlo (MC) explanations instead of amortized ones. However, this ignores potentially valuable information already computed by the amortized explainer. In this section, we propose a more effective solution called *explanations with initial guess*, which combines amortized and Monte Carlo explanations to improve quality.

**Explanation with Initial Guess** uses an optimized linear combination of the amortized explanation with a more computationally expensive method – the Monte Carlo explainer – to improve the quality of the explanation. We formally define *explanations with initial guess* next.

**Definition 2** (Explanation with Initial Guess). Given a Monte Carlo explainer $\mathsf{MC}^n(\boldsymbol{x}, \boldsymbol{y})$, and a combination function $\lambda_h : \mathbb{R}^d \to \mathbb{R}$ that reflects the quality of the amortized explanation $\mathsf{Amor}$, we define the explanation with initial guess as

$$\mathsf{IG}(\boldsymbol{x}, \boldsymbol{y}) \triangleq \lambda_h(\boldsymbol{x})\mathsf{Amor}(\boldsymbol{x}, \boldsymbol{y}) + (1 - \lambda_h(\boldsymbol{x}))\mathsf{MC}^n(\boldsymbol{x}, \boldsymbol{y}). \tag{8}$$

Recall that when $\tau_\alpha(\boldsymbol{x}) = 0$, selective explanations use the explanation with initial guess (1) to improve low-quality amortized explanations, i.e., $\mathsf{SE}(\boldsymbol{x}, \boldsymbol{y}) = \mathsf{IG}(\boldsymbol{x}, \boldsymbol{y})$.

Defining explanations with initial guess as the linear combination between the amortized and the Monte Carlo explanations is inspired by the literature on shrinkage estimators [21, 20] that use an initial guess ($\mathsf{Amor}(\boldsymbol{x}, \boldsymbol{y})$ in our case) to improve the estimation MSE in comparison to only using the empirical average (a role played by $\mathsf{MC}^n(\boldsymbol{x}, \boldsymbol{y})$ in our case). Next, we tune $\lambda_h$ to minimize the MSE from target explanations.

**Optimizing the Explanation Quality:** Our goal is for explanations with initial guess to approximate the target explanations, i.e., $||\mathsf{IG}(\boldsymbol{x}, \boldsymbol{y}) - \mathsf{Target}(\boldsymbol{x}, \boldsymbol{y})||$. To achieve this goal, we optimize the function $\lambda_h$ as follows.

First, since $\mathsf{Target}$ is unavailable, we use another Monte Carlo explanation $\mathsf{MC}^{n'}$ to approximate $\mathsf{Target}$. $\mathsf{MC}^{n'}$ is different from $\mathsf{MC}^n$ and potentially more computationally expensive but not necessarily. Importantly, $\mathsf{MC}^{n'}$ is only needed beforehand when computing $\lambda_h$, not at prediction time. In our experiments, we use SVS-12 for $\mathsf{MC}^{n'}$.

Second, we quantize the range of the uncertainty metric $s_h$ into bins to aggregate points with similar uncertainty and define the bins $Q_i$ by a partition $0 = \alpha_1 < \alpha_2 < ... < \alpha_m = 1$ of $[0, 1]$:

$$Q_i \triangleq [t_{\alpha_i}, t_{\alpha_{i+1}}), \ \forall i \in [m-1] \tag{9}$$

where $t_{\alpha_i}$ is defined as in (7). We then define the combination function to be

$$\lambda_h(\boldsymbol{x}) = \lambda_i \text{ if } s_h(\boldsymbol{x}) \in Q_i, \tag{10}$$

$\lambda_h$ is chosen to optimize the explanation-quality for points with similar uncertainty, $\lambda_i$ is given by:

$$\lambda_i \triangleq \underset{\lambda \in \mathbb{R}}{\operatorname{argmin}} \sum_{\substack{(\boldsymbol{x},\boldsymbol{y}) \in \mathcal{D}_{\mathtt{cal}} \\ s_h(\boldsymbol{x}) \in Q_i}} \left\| \mathsf{IG}(\boldsymbol{x},\boldsymbol{y}) - \mathsf{MC}^{n'}(\boldsymbol{x},\boldsymbol{y}) \right\|_2^2 . \tag{11}$$

The constant $\lambda_i$ is only computed once per bin and stored. At explanation time, when we provide explanations with initial guess (i.e., when $\tau_\alpha(\boldsymbol{x}) = 0$) (8), we lookup the bin for the point being explained and use the associated $\lambda_i$.

Theorem 1 provides a closed-form solution for $\lambda_i$.

**Theorem 1** (Optimal $\lambda_h$)**.** *Let $0 = \alpha_1 < \alpha_2 < ... < \alpha_m = 1$ and define $Q_i$ as in (9). Then the solution to the optimization problem in (11) is given by*

$$\lambda_i = \frac{\sum_{\substack{(\boldsymbol{x},\boldsymbol{y}) \in \mathcal{D}_{\mathtt{cal}} \\ s_h(\boldsymbol{x}) \in Q_i}} \langle \mathsf{MC}^n(\boldsymbol{x},\boldsymbol{y}) - \mathsf{MC}^{n'}(\boldsymbol{x},\boldsymbol{y}), \mathsf{MC}^n(\boldsymbol{x},\boldsymbol{y}) - \mathsf{Amor}(\boldsymbol{x},\boldsymbol{y}) \rangle}{\sum_{\substack{(\boldsymbol{x},\boldsymbol{y}) \in \mathcal{D}_{\mathtt{cal}} \\ s_h(\boldsymbol{x}) \in Q_i}} \|\mathsf{Amor}(\boldsymbol{x},\boldsymbol{y}) - \mathsf{MC}^n(\boldsymbol{x},\boldsymbol{y})\|_2^2} . \tag{12}$$

The range of uncertainty functions is **quantized** for two main reasons. First, the uncertainty metric $s_h$ encodes the amortized explainer's uncertainty for each point $\boldsymbol{x}$. This uncertainty quantification should be reflected in the choice of $\lambda_h$. Quantizing the range of $s_h$ allows us to group points with similar uncertainty levels and optimize $\lambda_h$ for each group separately. Second, quantizing the range of $s_h$ enables us to have multiple point per bin $Q_i$ allowing us to compute $\lambda_i$ to minimize the MSE in each bin.

We use the **Monte Carlo** explainer $\mathsf{MC}^{n'}$ because: (i) as mentioned above, we assume we don't have access to target explanations due to computational cost and (ii) even when using this Monte Carlo explainer, we show that in all bins, $\lambda_i$ approximates well the optimal combination function computed assuming access to target explanations from $\mathsf{Target}$ defined as

$$\lambda_i^{\mathtt{opt}} = \underset{\lambda \in [0,1]}{\operatorname{argmin}} \sum_{\substack{(\boldsymbol{x},\boldsymbol{y}) \in \mathcal{D}_{\mathtt{cal}} \\ s_h(\boldsymbol{x}) \in Q_i}} \|\mathsf{IG}(\boldsymbol{x},\boldsymbol{y}) - \mathsf{Target}(\boldsymbol{x},\boldsymbol{y})\|_2^2 .$$

Specifically, Theorem 2 shows that $\lambda_i \approx \lambda_i^{\mathtt{opt}}$ with high probability. Appendix E shows the formal version of the Theorem along with the proofs for all results in this section.

**Theorem 2** (**Informal** $\lambda_i \approx \lambda_i^{\mathtt{opt}}$)**.** *If (i) $\mathsf{MC}^n$ is sufficiently different from the amortized explainer $\mathsf{Amor}$ and (ii) $\mathsf{MC}^{n'}$ approximates the target explanations $\mathsf{Target}$ then $\lambda_i$ and $\lambda_i^{opt}$ are close with high-probability for all bins $Q_i$, i.e.,*

$$|\lambda_i - \lambda_i^{opt}| \le \epsilon \text{ with probability at least } 1 - e^{-C|Q_i|}.$$

*for a $C > 0$ and $|Q_i|$ is the number of points in the validation dataset $\mathcal{D}_{\mathtt{cal}}$ that are in bin $Q_i$.*

# 5    Experimental Results

This section analyzes the performance of selective explanations and its different components (i) uncertainty measures and (ii) explanations with initial guess. All results are showed in terms of MSE from target explanations, check Appendix D for the same results using Spearman's Rank Correlation.

**Experimental Setup:**    We generate selective explanations and evaluate their MSE and Spearman's correlation compared to the target explanation computed using a large number of inferences[1]. Although our results hold for any feature and data attribution method, in this section, we focus on Shapley values due to its frequent use and prevalence in the literature on amortized explainers [16, 6, 38]. Seaborn [36] is used to compute $95\%$ confidence intervals using the bootstrap method.

---

[1]We provide details on how target explanations were computed in Appendix D.1.

Table 1: Pearson's and Spearman's correlation between the proposed uncertainty measures and the MSE of amortized explanations from target SHAP explanations. Standard deviation in parenthesis.

| Correlation | Uncertainty Metric | Datasets | | | |
| --- | --- | --- | --- | --- | --- |
| | | UCI-Adult | UCI-News | Toxigen | Yelp |
| Pearson's | Deep | **0.37** (0.03) | **0.40** (0.03) | 0.54 (0.04) | 0.52 (0.04) |
| | Learned | 0.36 (0.03) | 0.18 (0.03) | **0.89** (0.02) | **0.72** (0.03) |
| Spearman's | Deep | 0.50 (0.03) | **0.43** (0.03) | 0.69 (0.03) | 0.55 (0.04) |
| | Learned | **0.69** (0.02) | 0.23 (0.03) | **0.93** (0.02) | **0.77** (0.03) |

**Datasets & Tasks:** We use four datasets: two tabular datasets UCI-Adult [1] and UCI-News [8], and two text classification datasets Yelp Review [41] and Toxigen [14]. We use 4000 samples from each dataset due to the cost of computing target explanations for evaluation. **Models:** For the tabular datasets, we train a multilayer perceptron [15] to learn the desired task. We use the HuggingFace Bert-based model `textattack/bert-base-uncased-yelp-polarity` [25] for the Yelp dataset and the Roberta-based model `tomh/toxigen_roberta` [14] for the Toxigen.[2] **Uncertainty metrics:** we train $k = 20$ amortized explainers per task to compute the deep uncertainty.

### 5.1 Uncertainty Measures & Explanations with Initial Guess

**Correlation Between MSE and Uncertainty Metrics:** Table 1 presents Pearson's and Spearman's correlation between the uncertainty metrics (Deep and Learned Uncertainty) and the MSE from amortized and target explanations. The table shows that the proposed uncertainty metrics positively correlate with the MSE, i.e., low uncertainty implies low MSE. Additionally, Spearman's correlation is specially high in the language models used in the Toxigen and Yelp datasets, our main object of interest. This finding indicates that the uncertainty metrics might perform especially well when detecting inaccurate amortized explanations attributed to the predictions of language models.

**Detecting High-MSE Explanations using Uncertainty:** Figure 3 presents the MSE of amortized explanations (y-axis) which are predicted to be higher-quality at a coverage level $\alpha$ (x-axis), i.e., MSE of points such that $\tau_\alpha(\boldsymbol{x}) = 1$. The Oracle[3] is computed by sorting examples from smallest to highest MSE – the optimal selection. The random selector chooses covered points uniformly at random. Figure 3 shows that both deep and learned uncertainty metrics successfully identify examples that receive lower and higher-accuracy amortized explanations, as also suggested by Table 1. Surprisingly, learned uncertainty can identify points that will receive low-accuracy amortized explanations almost as accurately as the optimal Oracle for the language models ((c) and (d)).

---

[2] For more details on implementation, please see Appendix D.1.

[3] The oracle is computationally expensive because it requires access to target explanations.

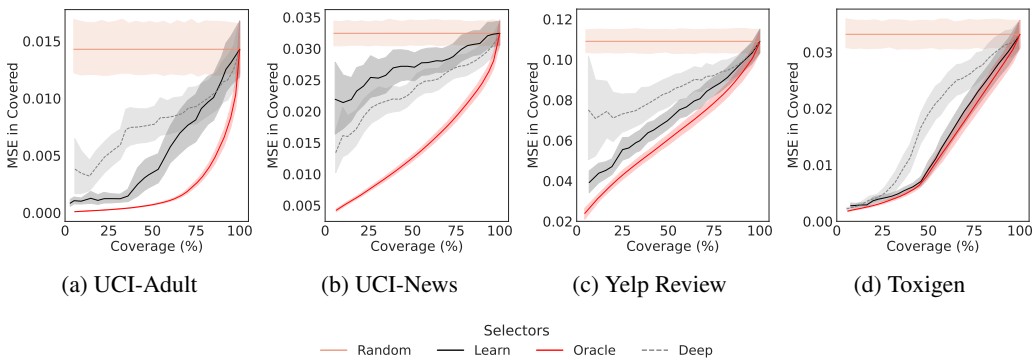

| (a) UCI-Adult | (b) UCI-News | (c) Yelp Review | (d) Toxigen |
| --- | --- | --- | --- |

Selectors
—— Random    —— Learn    —— Oracle    ---- Deep

Fig. 3: Coverage ($\alpha$) vs. MSE of covered points. A point $\boldsymbol{x}$ is covered when its amortized explanation is predicted to be higher-quality, i.e., $\tau_\alpha(\boldsymbol{x}) = 1$. When $\alpha = 100\%$ then all points are covered and the MSE of covered points is the average MSE for the amortized explainer.

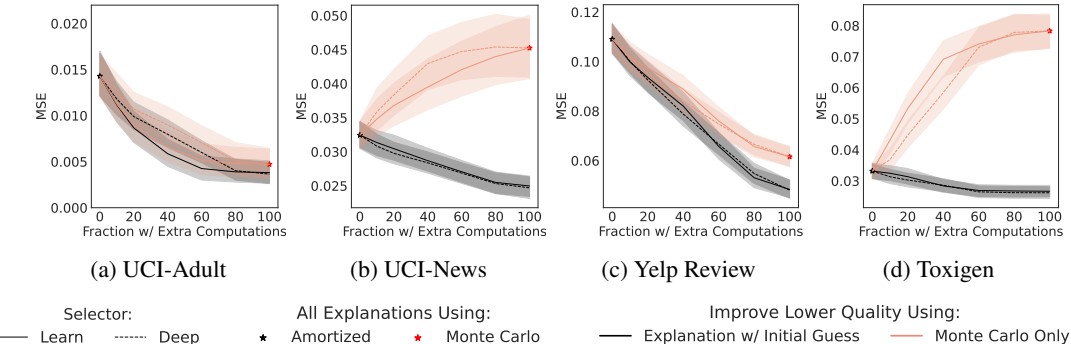

(a) UCI-Adult  (b) UCI-News  (c) Yelp Review  (d) Toxigen

Selector: —— Learn ······· Deep    All Explanations Using: ★ Amortized ★ Monte Carlo    Improve Lower Quality Using: —— Explanation w/ Initial Guess —— Monte Carlo Only

Fig. 4: Fraction $(1-\alpha)$ of points which explanations receive additional computations (x-axis) vs. MSE of selective explanations w.r.t. target explanations (y-axis) with coverage $\alpha$. Naive uses $\lambda_h = 0$ while Initial guess uses $\lambda_h$ in (12). MSE is computed across all points in the test dataset. Yelp Review and Toxigen use SVS (12) as Monte Carlo explanations while UCI-Adult and UCI-News use KS (32).

**Explanations with Initial Guess vs. Monte Carlo**    In Figure 4 we compare selective explanations improving quality of non-covered points using (i) explanations with initial guess and (ii) Monte Carlo explanations, when amortized explanations are inaccurate ($\lambda_h = 0$). **Case 1:** When the MSE from the Monte Carlo is smaller than from the amortized explainer ((a) and (c)), explanations with initial guess results in a smaller MSE compared to only using Monte Carlo. **Case 2:** When the MSE in Monte Carlo is larger than the amortized explanation MSE ((b) and (d)), only using Monte Carlo increase the MSE while explanations with initial guess reduces the MSE. Together, Cases 1 and 2 suggest that even when lower quality, explanations contain valuable information that can be leveraged by explanations with initial guess to improve explanation quality.

## 5.2 Efficacy of Selective Explanations

**Worst Case Performance Improvement:**    Figure 5 shows the MSE of selective explanations for the points receiving the highest MSEs. The figure suggests that selective explanations significantly decrease the worst-case MSE of amortized explanations. With just 20% coverage the MSE decreases consistently across datasets. Remarkably, when providing explanations with initial guess for 20% of the samples in the Yelp dataset (Figure 5 (c)), selective explanations result in MSE for the worst 5% of points that is about 30% smaller than the original amortized explanations – this is even more pronounced in the UCI datasets.

**Improved Inferences vs. Quality Trade-off:**    Figure 6 presents the trade-off between number of inferences per explanation and MSE from target explanations using selective and Monte Carlo explanations. The MSE decreases with the number of inferences and selective explanations Pareto dominates Monte Carlo explanations. We also show an "Oracle" that knows a priori how to optimally

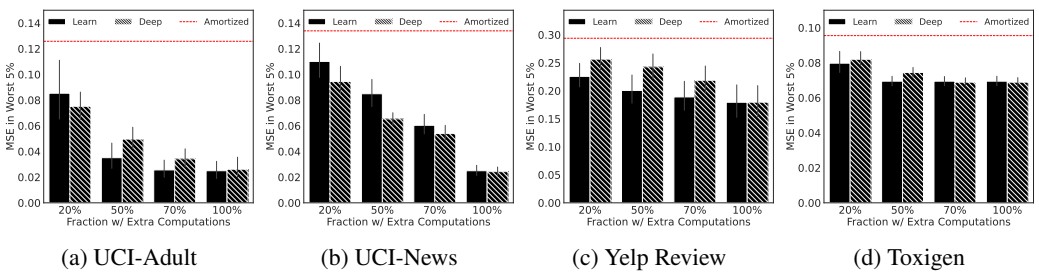

(a) UCI-Adult  (b) UCI-News  (c) Yelp Review  (d) Toxigen

Fig. 5: MSE for the 5% of explanations with the highest MSE in the test dataset (y-axis) for selective explanations with varying fraction of points with extra computations (x-axis). Selective explanations are shown in (i) black solid bar using the Learned uncertainty and (ii) striped black bar using Deep uncertainty. Dashed red line shows the MSE of amortized explanations in worst 5% explanations.

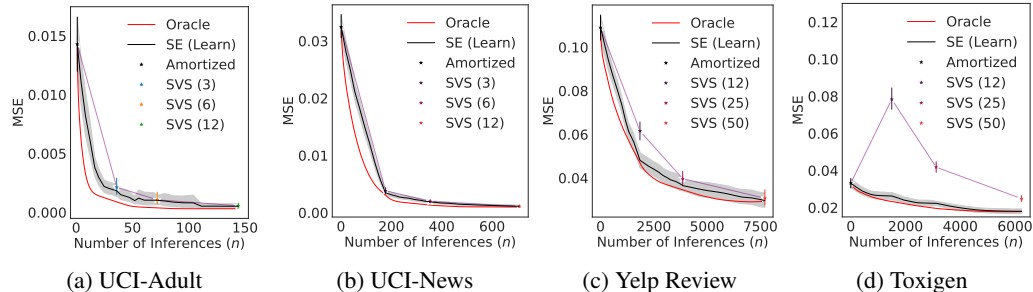

| (a) UCI-Adult | (b) UCI-News | (c) Yelp Review | (d) Toxigen |

Fig. 6: Number of inferences (x-axis) vs. MSE (y-axis). Black curve shows the performance of selective explanations using Learned uncertainty. Purple curve connects Shapley Value Sampling (SVS) with parameters 12, 25, and 50 sequentially until all samples receive SVS-50 explanations and amortized explanations. The red curve is a the Oracle that optimally trades off MSE and inferences.

route samples in terms of MSE and inferences. We simulate this oracle by pre-computing SVS explanations with parameters 12, 25, and 50, and selecting the one with the smallest MSE from the target SHAP explanation while manteining the average number inferences shown in x-axis.

Remarkably, Figure 6 shows that selective explanations closely approximate the Oracle curve, indicating that, on these benchmarks, our method has a near-oracle trade-off between the number of inferences and MSE.

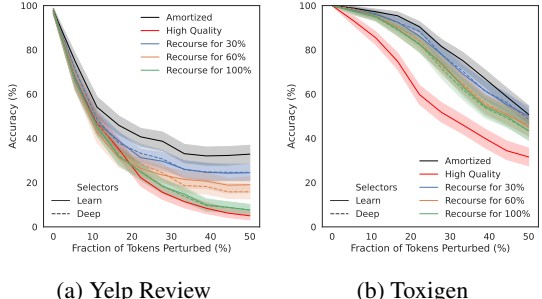

| (a) Yelp Review | (b) Toxigen |

Fig. 7: Model accuracy (y-axis) when removing the tokens with the highest attribution scores according to the amortized explainer (black), selective explanations with varying coverage and target explanations (red).

**Improved Local Fidelity:** Figure 7 shows that selective explanations increase the local fidelity of the amortized explainer and that local fidelity increases with the fraction of points that receive additional computations (recourse). Both Yelp and Toxigen models receive explanations with initial guess using SVS-12.

## 6 Final Remarks

**Conclusion:** We propose *selective explanations* that first identify which inputs would receive a low-quality but computationally cheap explanation (amortized) and then perform additional model inferences to improve the quality of these explanations. We propose *explanations with initial guess* to improve the quality of explanations by combining amortized explanations with more expensive explanations Monte Carlo using an optimized combination function, improving the explanation performance. Selective explanations provide a new framework for approximating expensive feature attribution methods. Our experiments indicate that selective explanations (i) efficiently identify points that the amortized explainer would produce low-quality explanations, (ii) improves the quality of the worst-quality amortized explanations, (iii) improves the trade-off between computational cost and explanation quality, and (iv) improves the local fidelity of amortized explanations.

**Limitations:** Selective explanations can be applied to any feature attribution method for which amortized and Monte Carlo explainers were developed. However, our empirical results focus on Shapley values. We leave the application of selective explanations to other attribution methods for future work. Additionally, we do not explore image classifiers, which may also interest the interpretability community. Also, we do not explore selective explanations for Generative Language models due to the lack of amortized explainers for such application.

# 7 Acknowledgments

The authors thank Amit Dhurandhar for early discussions on the trustworthiness of amortized explainers. This material is based upon work supported by the National Science Foundation under awards CAREER-1845852, CIF-1900750, CIF-2231707, and CIF-2312667, FAI-2040880, and also an Apple Scholar Fellowship. The views expressed here are those of the authors and do not reflect the official policy or position of the funding agencies.

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

# A   Overview

In this supplementary material we provide the following information:

- Appendix B discuss other high-quality and Monte Carlo explainers.
- Appendix C discuss a guide to select the coverage $\alpha$ when the agent providing selective explanations has a budget for the average number of inferences to provide an explanation.
- Appendix D shows more experimental results on selective explanations.
- Appendix E shows the proofs for the theoretical results in Section 4.

# B   Additional Explanation Methods

In this section, we describe high-quality, Monte Carlo, and amortized explainers with further details.

## B.1   High-Quality Explainers

**Shapley Values (SHAP)** [22] is a **high-quality** explainer that attributes a value $\phi_i$ for each feature $x_i$ in $\boldsymbol{x} = (x_1, ..., x_d)$ which is the marginal contribution of feature $x_i$ if the model was to predict $\boldsymbol{y}$ (2).

$$\phi_i(\boldsymbol{x}, \boldsymbol{y}) = \frac{1}{d} \sum_{S \subset [d]/\{i\}} \binom{d-1}{|S|}^{-1} \left( h_{\boldsymbol{y}}(\boldsymbol{x}_{S \cup \{i\}}) - h_{\boldsymbol{y}}(\boldsymbol{x}_S) \right). \tag{13}$$

SHAP has several desirable properties and is widely used. However, as (2) indicates, computing Shapley values and the attribution vector $\mathsf{Target}(\boldsymbol{x}, \boldsymbol{y}) = (\phi_1(\boldsymbol{x}, \boldsymbol{y}), ..., \phi_d(\boldsymbol{x}, \boldsymbol{y}))$ requires $2^d$ inferences from $h$, making SHAP impractical for large models where inference is costly. This has motivated several approximation methods for SHAP, discussed next[4].

**Local Interpretable Explanations (Lime).**   Lime is another feature attribution method [29] widely used to provide feature attributions. It relies on selecting combinations of features, removing these features from the input to generate perturbations, and using these perturbations to approximate the black box model $h$ locally by a linear model. The coefficients of the linear model are considered to be the attribution of each feature. Formally, given a weighting kernel $\pi(S)$ and a penalty function $\Omega$, the attribution produced by lime are given by

$$(\phi, a) = \operatorname*{argmin}_{\phi \in \mathbb{R}^d, a \in \mathbb{R}} \sum_{S \subset [d]} \pi(S) \left( h(\boldsymbol{x}_S) - a_0 - \sum_{i \in S} \phi_i \right), \tag{14}$$

where $\mathsf{Target}(\boldsymbol{x}, \boldsymbol{y}) = \phi$. As in SHAP, to compute the feature attributions using lime, we need to perform a large number of model inferences, which is prohibitive for large models.

## B.2   Monte Carlo Lime

**Shapley Value Sampling (SVS)** [24] is a **Monte Carlo** explainer that approximates SHAP by restricting the sum in (2) to specific permutations of feature. SVS computes the attribution scores by uniformly sampling $m$ features permutations $S_1, ..., S_m$ restricting the sum in (2) and performing $n = md + 1$ inferences. We denote SVS that samples $m$ feature permutations by SVS-$m$.

**Kernel Shap (KS)** [22] is a **Monte Carlo** explainer that approximate the Shapley values using the fact that SHAP can be computed by solving the optimization problem

$$(\phi, a) = \operatorname*{argmin}_{\phi \in \mathbb{R}^d, a \in \mathbb{R}} \sum_{i=1}^{n} \pi(S_i) \left( h(\boldsymbol{x}_{S_i}) - a_0 - \sum_{j \in S_i} \phi_j \right), \tag{15}$$

using $\pi(S) = \binom{d}{|S|} |S|(d - |S|)$ and where $\mathsf{MC}^n(\boldsymbol{x}, \boldsymbol{y}) = \phi$. Kernel Shap samples $n > 0$ feature combinations $S_1, ..., S_n$ and define the feature attributions to be given by the coefficients $\phi$. We refer to Kernel Shap using $n$ inferences as KS-$n$. We use the KS-$n$ from the Captum library [18] for our experiments.

---

[4]We also discuss Lime and its amortized version in Appendix B

**Sample Constrained Lime.** To approximate the attributions from Lime, we consider the sample-contained version of (15). Instead of sampling all feature combinations in $[d]$, we only uniformly sample a fixed number $n$ of feature combinations $S_1, ..., S_n$. For our experiments, shown in the appendix, we use the Sample Constrained Lime from the Captum library [18].

### B.3 Amortized Explainers

**Stochastic Amortization** [6] is a **Amortized** explainer that uses noisy Monte Carlo explanations to learn high-quality explanations. Covert et al. [6] trained an amortized explainer $\mathsf{Amor} \in \mathcal{F}$ in a hypothesis class $\mathcal{F}$ (we use multilayer perceptrons) that takes an input and predicts an explanation. Specifically, taking the amortized explainer to be the solution of the training problem given in (3).

$$\mathsf{Amor} \in \underset{f \in \mathcal{F}}{\arg\min} \sum_{(\boldsymbol{x}, \boldsymbol{y}) \in \mathcal{D}_{\text{train}}} \| f(\boldsymbol{x}, \boldsymbol{y}) - \mathsf{MC}^n(\boldsymbol{x}, \boldsymbol{y}) \|_2^2. \tag{16}$$

We are interested in explaining the predictions of large models for text classification. However, the approach in (3) is only suitable for numerical inputs. Hence, we follow the approach from Yang et al. [38] to explain the predictions of large language models, explained next.

**Amortized Shap for LLMs** [38] is a **Amortized** explainer similar to the one in (3) but tailored for LLMs. First, the authors note that they can use the LLM to write all input texts $\boldsymbol{x}$ as a sequence of token embedding $[e_1(\boldsymbol{x}), ..., e_{|\boldsymbol{x}|}(\boldsymbol{x})]$ where $e_i(\boldsymbol{x}) \in \mathbb{R}^d$ denotes the LLM embedding for the $i$-th token contained in the input text $\boldsymbol{x}$ and $|\boldsymbol{x}|$ is the number of tokens in the input text. Second, they restrict $\mathcal{F}$ in (3) to be the set of all linear regressions that take the token embeddings and output the token attribution score. Then, they solve the optimization problem in

$$W \in \underset{W \in \mathbb{R}^d, b \in \mathbb{R}}{\arg\min} \sum_{(\boldsymbol{x}, \boldsymbol{y}) \in \mathcal{D}_{\text{train}}} \sum_{j=1}^{|\boldsymbol{x}|} \| W^T e_j(\boldsymbol{x}) + b - \mathsf{MC}^n(\boldsymbol{x}, \boldsymbol{y})_j \|_2^2, \tag{17}$$

and define the amortized explainer as $\mathsf{Amor}(\boldsymbol{x}) = (W^T e_1(\boldsymbol{x}) + b, ..., W^T e_{|\boldsymbol{x}|}(\boldsymbol{x}) + b)$.

We use stochastic amortization to produce amortized explainers for tabular datasets and Amortized Shap for LLMs to produce explainers for LLM predictions. Both explainers are trained using SVS-12 as $\mathsf{MC}^n$.

## C Selecting Coverage for a Given Inference Budget

**Determining Coverage from Inference Budget:** Providing explanations with initial guess increases the number of model inferences from 1 when using solely the amortized explainer to $n + 1$. However, a practitioner may have a budget of inferences, i.e., a maximum average number of inferences they are willing to perform to provide an explanation. We formalize the notion of inference budget in Definition 3.

**Definition 3** (Inference Budget). Denote by $\mathsf{N}(\mathsf{SE}(\boldsymbol{x}, \boldsymbol{y}))$ the number of model inferences to produce the explanation $\mathsf{SE}(\boldsymbol{x}, \boldsymbol{y})$. The inference budget $\mathsf{N}_{\text{budget}} \in \mathbb{N}$ is the maximum average number of inferences a practitioner is willing to perform per explanation, i.e., it is such that

$$\mathsf{N}_{\text{budget}} \geq \mathbb{E}\left[\mathsf{N}(\mathsf{SE}(\boldsymbol{x}, \boldsymbol{y}))\right]. \tag{18}$$

Once an inference budget $\mathsf{N}_{\text{budget}}$ is defined, the coverage $\alpha$ should be set to follow it. In Proposition 1, we show the minimum coverage for the selective explanations to follow the inference budget.

**Proposition 1** (Coverage for Inference Budget). *Let $N_{budget} \geq 1$ be the inference budget, and assume that the Monte Carlo method $MC^n(\boldsymbol{x}, \boldsymbol{y})$ uses $n$ model inferences. Then, the coverage level $\alpha$ should be chosen such that*

$$\frac{n + 1 - N_{budget}}{n} = \min_{\alpha \in [0,1]} \alpha, \text{ such that } \mathbb{E}\left[N(\mathsf{SE}(\boldsymbol{x}, \boldsymbol{y}))\right] \leq N_{budget}. \tag{19}$$

*Recall that SVS-$m$ performs $n = 1 + dm$ inferences ($\boldsymbol{x} \in \mathbb{R}^d$), and KS-$m$ performs $n = m$ inferences.*

# D More Experimental Results

In this section, we (i) give further implementation details and (ii) discuss further empirical results.

## D.1 More Details on Experimental Setup

**High-Quality Explanations:** We define the high-quality explanations for the tabular datasets to be given by Kernel Shap with as many inferences as needed for convergence, using the Shapley Regression library [4]. For the textual dataset, following [38], we define the high-quality explanations to be given by Kernel Shap using $8912$ model inferences per explanation.

**Amortized Explainers:** For the tabular datasets, we use the amortized explainer from [6] that we describe in Section 2. Specifically, we use a multilayer perceptrom model architecture to learn the shapley values for the tabular datasets. For the textual datasets, we use the linear regression on token-level textual embeddings to learn the shapley values, as described in Section 2. Both amortized models learn from the training dataset of explanations generated using Shapley Value Sampling from the Captum library [18] with parameter 12, i.e., SVS-12.

**Uncertainty Metrics:** We test the two proposed uncertainty metrics in Section 3, namely, deep uncertainty and uncertainty learn. For **deep uncertainty**, we run the training pipeline for the amortized explainers 20 times for each dataset we perform experiments on, resulting in 20 different amortized explainer that we use to compute (4). For **uncertainty learn**, we use the multilayer perceptrom as the hypothesis class with only one hidden layer. The hidden layer was composed of $\kappa = 3d$ neurons where $d$ is the dimension of the input vector $x \in \mathbb{R}^d$. The uncertainty learn metric was trained on $\mathcal{D}_{\texttt{train}}$, the same training dataset as the amortized explainers.

**Dataset sizes:** We use 4000 samples from each dataset due to computational limitations on the computation of high-quality explanations used to evaluate selective explanations. All explanations were computed using the Captum library [18]. The dataset $\mathcal{D}$ with $N = 4000$ samples was partitioned in three parts, $\mathcal{D}_{\texttt{train}}$ with $50\%$ of points, $\mathcal{D}_{\texttt{cal}}$ with $25\%$ of points, and $\mathcal{D}_{\texttt{test}}$ with the other $25\%$ of points.

**Computational Resources:** All experiments were run in a A100 40 GB GPU. For each dataset, we compute different Monte Carlo explanations. For the UCI-News dataset, the high quality explanations took 4:30 hours to be generate until convergence while for UCI-Adult it took 3:46 hours. For the tabular datasets, all other Monte Carlo explainers were generated in less than 1 hour. For the language models, the high-quality explanations with 8192 model inferences, took 18:51 hours for the Toxigen dataset and 20:00 hours for the Yelp Review datasets. The other used Monte Carlo explanations took proportional (to the number of inferences) time to be generated.

## D.2 Uncertainty Measures Impact on Spearman's Correlation

Figure 8 shows in the x-axis the coverage ($\alpha$) and in the y-axis the average Spearman's correlation of the selected amortized explanations from high-quality explanations using deep uncertainty (with 20 models) and the uncertainty learn to select low-quality explanations. The Oracle[5] is computed by sorting examples by the smallest to higher MSE and computing the average Spearman's correlation in the bottom x-axis points accordingly to the MSE and is the best that can be done in terms of MSE.

Figure 8 shows that the Oracle and proposed uncertainty metrics don't always select the points with the smallest Spearman's correlation first. This implies that MSE and Spearman's correlation don't always align, i.e., there are points with high MSE and high Spearman's correlation at the same time. However, we note that the uncertainty learns selector can be applied to **any** metric $\ell$ as we define in (5) including Spearman's correlation and any combination of Spearman's correlation and MSE aiming to approximate both metrics. Moreover, when the smallest MSE aligns with the highest Spearman's correlation, i.e., the oracle is decreasing in Spearman's correlation when the coverage increases (Figure 8 (a) and (c)), the proposed uncertainty metrics also accurately detect the low-quality explanations in term of Spearman's correlation.

---

[5]The oracle is computationally expensive because it requires access to high-quality explanations.

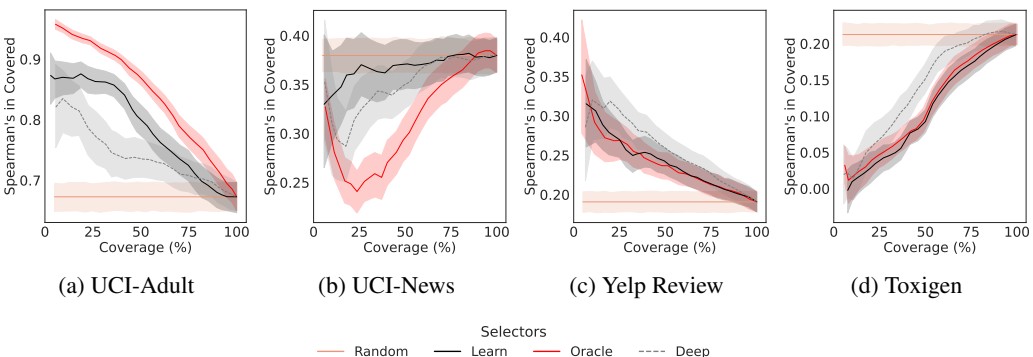

Selectors
—— Random  —— Learn  —— Oracle  ---- Deep

| (a) UCI-Adult | (b) UCI-News | (c) Yelp Review | (d) Toxigen |

Fig. 8: Coverage vs. Spearman's correlation from the high-quality explanation. Coverage is the percentage of the points that the selection function predicts that will receive a higher-quality explanation, i.e., $\tau_t(\boldsymbol{x}) = 1$. When coverage is $100\%$ Spearman's correlation is the average performance for the amortized explainer.

## D.3  The Effect of Explanations with Initial Guess

In Figure 9 we compare explanations with initial guess (Definition 2) to only using the Monte Carlo to provide recourse to the low-quality explanaitons, i.e., $\lambda_h = 0$ we call it Naive. In all tested cases, Spearman's correlation of the Monte Carlo method is comparable to or larger than the amortized explainer. Although selective explanations optimized for MSE by using explanations with initial guess (Definition 2), we observe that the Spearman's correlation of selective explanations is close to or larger than the naive method, once again, demonstrating the efficacy of selective explanations.

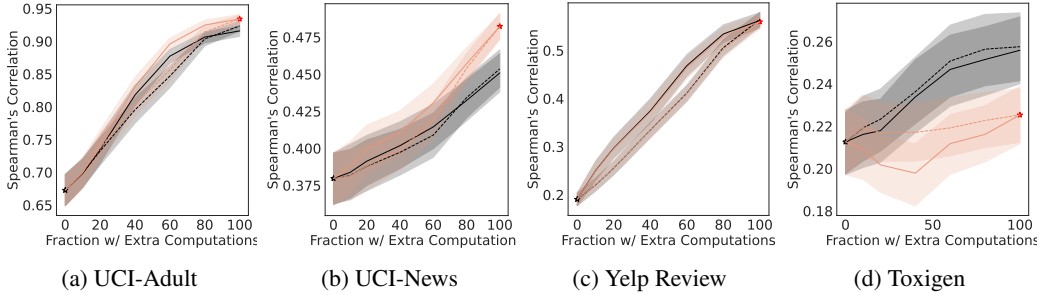

| (a) UCI-Adult | (b) UCI-News | (c) Yelp Review | (d) Toxigen |

Fig. 9: Fraction of the population that receive explanations with initial guess (x-axis) vs. their Spearman's correlation from the high-quality explanations (y-axis). Naive uses $\lambda_h = 0$ while initial guess uses explanations with initial guess, i.e., when $\lambda_h$ is given in (12).

## D.4  Spearman's correlation Explanation of initial guess in the worst explanations

Figure 10 shows the Spearman's rank correlation of selective explanations for the points receiving explanations with the smallest correlation. The figure shows that selective explanations significantly decrease the worst-case Spearman's rank correlation of amortized explanations. With just $20\%$ coverage the Spearman's rank correlation increases consistently across datasets. Remarkably, when providing explanations with initial guess for $50\%$ of the samples in the Yelp dataset (Figure 10 (c)), selective explanations result in explanations with positive correlation with target explanations in the worst $5\%$. At the same worst $5\%$ of points, amortized explanations are negatively correlated with target explanations.

## D.5  Performance for Different Monte-Carlo Explainers

Figure 11 shows how the MSE and Spearman's correlation behave accordingly with the quality of the Monte Carlo explainer. We compare Kernel Shap and Shapley Value Sampling in all experiments. We observe that when the quality of the Monte Carlo explainer increases, the quality of the Selective explanation also increases, i.e., the MSE decreases and the Spearman's correlation increases. Moreover,

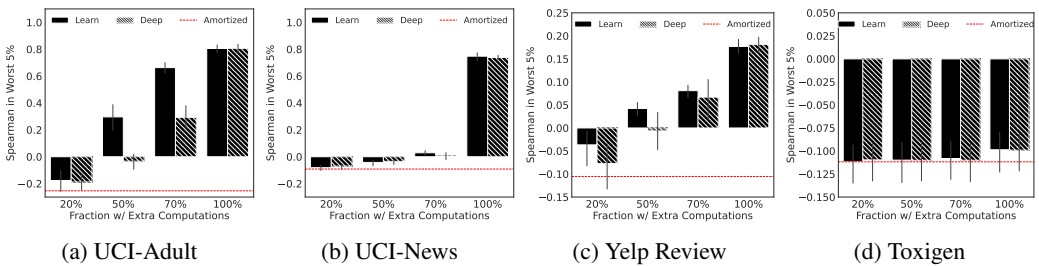

(a) UCI-Adult      (b) UCI-News      (c) Yelp Review      (d) Toxigen

Fig. 10: Spearman's correlation for the 5% of explanations with the smallest correlation in the test dataset (y-axis) for selective explanations with varying fraction of points with extra computations (x-axis). Selective explanations are shown in (i) black solid bar using the Learned uncertainty and (ii) striped black bar using Deep uncertainty. Dashed red line shows the MSE of amortized explanations in worst 5% explanations.

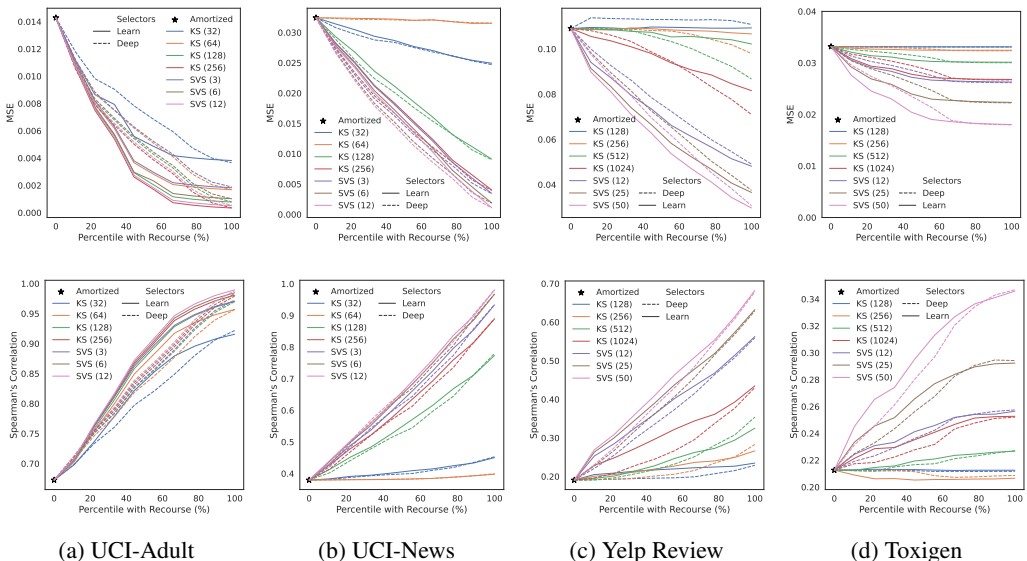

(a) UCI-Adult      (b) UCI-News      (c) Yelp Review      (d) Toxigen

Fig. 11: MSE (top) and Spearman's correlation (bottom) for selective explanations using different Monte Carlo explainers.

we also observe diminishing returns, i.e., after a certain point, increasing the quality of the Monte Carlo explanations doesn't lead to a tailored increase in performance. For example, observe the SVS method in the tabular datasets Figure 11 (a) and (b). We also observe that providing explanations with initial guess has a high impact on both Spearman's correlation and MSE when only providing recourse toa small fraction of the population. For example, when providing explanations with initial guess for 20% of the population using SVS-12 in the Yelp Review dataset, Figure 11 (c), increases the Spearman's correlation in more than 50% (from 0.2 to more than 0.3).

### D.6  Time Sharing Using Selective Explanations

Figure 6 presents the trade-off between number of inferences per explanation and MSE from target explanations using selective and Monte Carlo explanations. In addition to what is shown in Figure 6, we also show selective explanations using the Deep uncertainty metric as a selector.

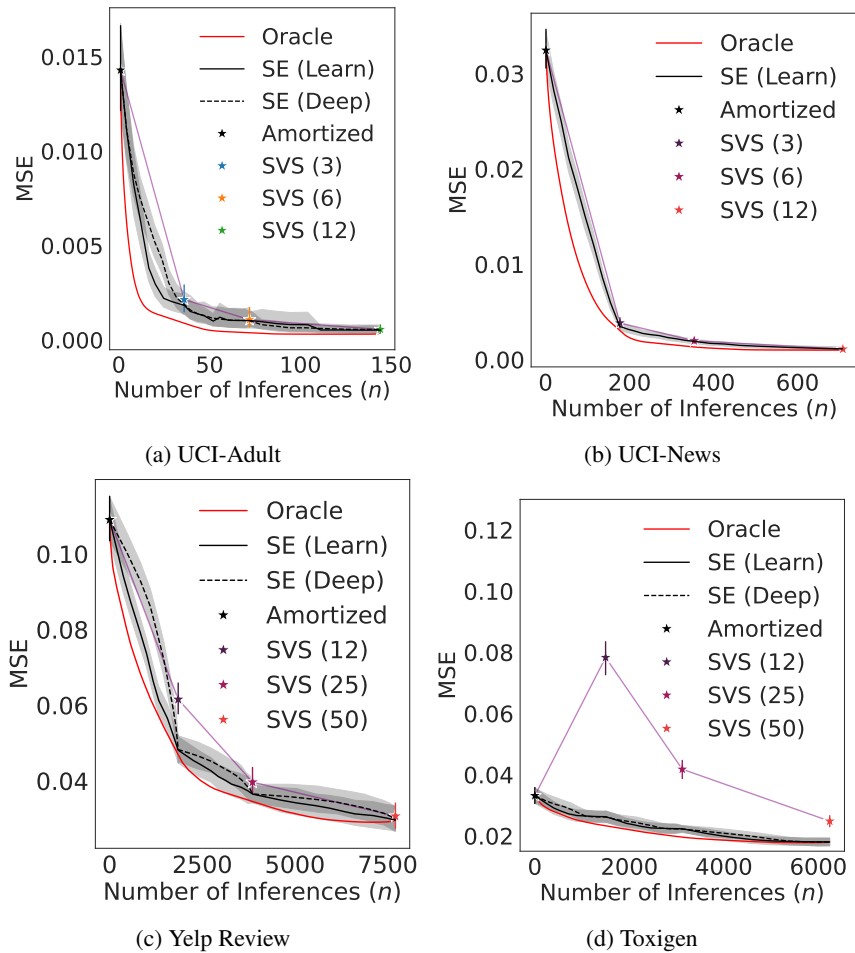

(a) UCI-Adult

(b) UCI-News

(c) Yelp Review

(d) Toxigen

Fig. 12: Number of inferences (x-axis) vs. MSE (y-axis). Black curve shows the performance of selective explanations using Learned uncertainty. Purple curve connects Shapley Value Sampling (SVS) with parameters 12, 25, and 50 sequentially until all samples receive SVS-50 explanations and amortized explanations. The red curve is a the Oracle that optimally trades off MSE and inferences

## E Proofs of Theoretical Results

**Theorem 1** (Optimal $\lambda_h$). *Let $0 = \alpha_1 < \alpha_2 < ... < \alpha_m = 1$ and define $Q_i$ as in* (9). *Then, $\lambda_i$ that solves the optimization problem in* (11) *is given by*

$$\lambda_i = \frac{\sum_{\substack{(\boldsymbol{x},\boldsymbol{y})\in\mathcal{D}_{val} \\ s_h(\boldsymbol{x})\in Q_i}} \langle \mathsf{MC}^n(\boldsymbol{x},\boldsymbol{y}) - \mathsf{MC}^{n'}(\boldsymbol{x},\boldsymbol{y}), \mathsf{MC}^n(\boldsymbol{x},\boldsymbol{y}) - \mathsf{Amor}(\boldsymbol{x},\boldsymbol{y}) \rangle}{\sum_{\substack{(\boldsymbol{x},\boldsymbol{y})\in\mathcal{D}_{val} \\ s_h(\boldsymbol{x})\in Q_i}} ||\mathsf{Amor}(\boldsymbol{x},\boldsymbol{y}) - \mathsf{MC}^n(\boldsymbol{x},\boldsymbol{y})||_2^2}. \tag{20}$$

*Proof.* First, recall that

$$\lambda_i \triangleq \operatorname*{argmin}_{\lambda\in\mathbb{R}} \sum_{\substack{(\boldsymbol{x},\boldsymbol{y})\in\mathcal{D}_{val} \\ s_h(\boldsymbol{x})\in Q_i}} \left\lVert \mathsf{SE}(\boldsymbol{x},\boldsymbol{y}) - \mathsf{MC}^{n'}(\boldsymbol{x},\boldsymbol{y}) \right\rVert_2^2 \tag{21}$$

$$= \operatorname*{argmin}_{\lambda\in\mathbb{R}} \sum_{\substack{(\boldsymbol{x},\boldsymbol{y})\in\mathcal{D}_{val} \\ s_h(\boldsymbol{x})\in Q_i}} \left\lVert \lambda\mathsf{Amor}(\boldsymbol{x},\boldsymbol{y}) + (1-\lambda)\mathsf{MC}^n(\boldsymbol{x},\boldsymbol{y}) - \mathsf{MC}^{n'}(\boldsymbol{x},\boldsymbol{y}) \right\rVert_2^2. \tag{22}$$

Note that the function in (22) is convex in $\lambda$; therefore, if the derivative of it with respect to $\lambda$ is zero, then the lambda that achieves the zero gradient is the minima. So, let's derivate (22) to find $\lambda_i$.

$$0 = \frac{d}{d\lambda} \sum_{\substack{(\boldsymbol{x},\boldsymbol{y})\in\mathcal{D}_{\mathtt{val}} \\ s_h(\boldsymbol{x})\in Q_i}} \left\|\lambda\mathsf{Amor}(\boldsymbol{x},\boldsymbol{y}) + (1-\lambda)\mathsf{MC}^n(\boldsymbol{x},\boldsymbol{y}) - \mathsf{MC}^{n'}(\boldsymbol{x},\boldsymbol{y})\right\|_2^2 \tag{23}$$

$$= 2 \sum_{\substack{(\boldsymbol{x},\boldsymbol{y})\in\mathcal{D}_{\mathtt{val}} \\ s_h(\boldsymbol{x})\in Q_i}} \lambda||\mathsf{MC}^n(\boldsymbol{x},\boldsymbol{y}) - \mathsf{Amor}(\boldsymbol{x},\boldsymbol{y})||^2 \tag{24}$$

$$- 2 \sum_{\substack{(\boldsymbol{x},\boldsymbol{y})\in\mathcal{D}_{\mathtt{val}} \\ s_h(\boldsymbol{x})\in Q_i}} \langle\mathsf{MC}^n(\boldsymbol{x},\boldsymbol{y}) - \mathsf{MC}^{n'}(\boldsymbol{x},\boldsymbol{y}), \mathsf{MC}^n(\boldsymbol{x},\boldsymbol{y}) - \mathsf{Amor}(\boldsymbol{x},\boldsymbol{y})\rangle \tag{25}$$

From (25) we conclude the proof by showing that

$$\lambda_i = \lambda = \frac{\sum_{\substack{(\boldsymbol{x},\boldsymbol{y})\in\mathcal{D}_{\mathtt{val}} \\ s_h(\boldsymbol{x})\in Q_i}} \langle\mathsf{MC}^n(\boldsymbol{x},\boldsymbol{y}) - \mathsf{MC}^{n'}(\boldsymbol{x},\boldsymbol{y}), \mathsf{MC}^n(\boldsymbol{x},\boldsymbol{y}) - \mathsf{Amor}(\boldsymbol{x},\boldsymbol{y})\rangle}{\sum_{\substack{(\boldsymbol{x},\boldsymbol{y})\in\mathcal{D}_{\mathtt{val}} \\ s_h(\boldsymbol{x})\in Q_i}} ||\mathsf{MC}^n(\boldsymbol{x},\boldsymbol{y}) - \mathsf{Amor}(\boldsymbol{x},\boldsymbol{y})||^2}. \tag{26}$$

$\square$

**Theorem 2** ($\lambda_i \approx \lambda_i^{\mathtt{opt}}$). *Let the Monte Carlo explanation used to provide recourse $\mathsf{MC}^n$ to be different enough from the amortized explainer, i.e., $\mathbb{E}\left[||\mathsf{MC}^n(X,Y) - \mathsf{Amor}(X,Y)||^2\right] = \mu > 0$. Also, assume that $\mathsf{MC}^{n'}$ is a good Monte Carlo approximation for the high-quality explainer $\mathsf{Target}$, i.e., $\mathbb{E}\left[||\mathsf{MC}^{n'}(X,Y) - \mathsf{Target}(X,Y)||^2\right] = \mu^*$ for $\epsilon > \frac{\sqrt{5\mu^*}}{\mu}$. Recall that $\boldsymbol{x} \in \mathbb{R}^d$. If the explanations are bounded, i.e., $||\mathsf{MC}^n(\boldsymbol{x},\boldsymbol{y})||, ||\mathsf{Amor}(\boldsymbol{x},\boldsymbol{y})||, ||\mathsf{Target}(\boldsymbol{x},\boldsymbol{y})| < Cd$ for some $C > 0$ then*

$$\Pr[|\lambda_i - \lambda_i^{opt}| > \epsilon] \leq e^{\frac{-\mu^2|Q_i|}{4Cd}} + e^{\frac{-\mu^4\epsilon^4|Q_i|}{400Cd}}, \tag{27}$$

*where $|Q_i|$ is the number of points $\boldsymbol{x}$ in the validation dataset $\mathcal{D}_{\mathtt{val}}$ that are in the bin $Q_i$.*

*Proof.* Denote $|Q_i| = |\{(\boldsymbol{x},\boldsymbol{y}) \in \mathcal{D}_{\mathtt{val}}, \text{ s.t. } s_h(\boldsymbol{x}) \in Q_i\}|$.

We start by showing that if $\mathbb{E}\left[||\mathsf{MC}^n(X,Y) - \mathsf{Amor}(X,Y)||^2\right] = \mu$ then

$$\Pr\left[\frac{1}{|Q_i|} \sum_{\substack{(\boldsymbol{x},\boldsymbol{y})\in\mathcal{D}_{\mathtt{val}} \\ s_h(\boldsymbol{x})\in Q_i}} ||\mathsf{MC}^n(\boldsymbol{x},\boldsymbol{y}) - \mathsf{Amor}(\boldsymbol{x},\boldsymbol{y})||^2 \leq \frac{\mu}{2}\right] \tag{28}$$

$$= \Pr\left[\mu - \frac{1}{|Q_i|} \sum_{\substack{(\boldsymbol{x},\boldsymbol{y})\in\mathcal{D}_{\mathtt{val}} \\ s_h(\boldsymbol{x})\in Q_i}} ||\mathsf{MC}^n(\boldsymbol{x},\boldsymbol{y}) - \mathsf{Amor}(\boldsymbol{x},\boldsymbol{y})||^2 \geq \frac{\mu}{2}\right] \tag{29}$$

$$\leq e^{\frac{-\mu^2|Q_i|}{4Cd}}. \tag{30}$$

Where the inequality in (30) follows from Hoeffding's inequality and the fact that:

$$||\mathsf{MC}^n(\boldsymbol{x},\boldsymbol{y}) - \mathsf{Amor}(\boldsymbol{x},\boldsymbol{y})||^2 \leq ||\mathsf{MC}^n(\boldsymbol{x},\boldsymbol{y})|| + ||\mathsf{Amor}(\boldsymbol{x},\boldsymbol{y})|| \leq 2Cd. \tag{31}$$

Second, we recall that $\mathbb{E}\left[||\mathsf{MC}^{n'}(X,Y) - \mathsf{Target}(X,Y)||^2\right] = \mu^* \leq \frac{\mu^2\epsilon^2}{5}$. Then, we have that

$$\Pr\left[\frac{1}{|Q_i|} \sum_{\substack{(\boldsymbol{x},\boldsymbol{y})\in\mathcal{D}_{\mathtt{val}} \\ s_h(\boldsymbol{x})\in Q_i}} ||\mathsf{Target}(\boldsymbol{x},\boldsymbol{y}) - \mathsf{Amor}(\boldsymbol{x},\boldsymbol{y})||^2 \geq \epsilon^2\frac{\mu^2}{4}\right] \tag{32}$$

$$= \Pr\left[\frac{1}{|Q_i|} \sum_{\substack{(\boldsymbol{x},\boldsymbol{y})\in\mathcal{D}_{\text{val}} \\ s_h(\boldsymbol{x})\in Q_i}} ||\mathsf{Target}(\boldsymbol{x},\boldsymbol{y}) - \mathsf{Amor}(\boldsymbol{x},\boldsymbol{y})||^2 - \mu^* \geq \epsilon^2 \frac{\mu^2}{4} - \mu^*\right] \tag{33}$$

$$\leq \Pr\left[\frac{1}{|Q_i|} \sum_{\substack{(\boldsymbol{x},\boldsymbol{y})\in\mathcal{D}_{\text{val}} \\ s_h(\boldsymbol{x})\in Q_i}} ||\mathsf{Target}(\boldsymbol{x},\boldsymbol{y}) - \mathsf{Amor}(\boldsymbol{x},\boldsymbol{y})||^2 - \mu^* \geq \epsilon^2 \frac{\mu^2}{20}\right] \tag{34}$$

$$\leq e^{\frac{-\mu^4 \epsilon^4 |Q_i|}{400 C d}}. \tag{35}$$

Where the inequality in (35) follows from Hoeffding's inequality and the fact that:

$$||\mathsf{Target}(\boldsymbol{x},\boldsymbol{y}) - \mathsf{Amor}(\boldsymbol{x},\boldsymbol{y})||^2 \leq ||\mathsf{Target}(\boldsymbol{x},\boldsymbol{y})|| + ||\mathsf{Amor}(\boldsymbol{x},\boldsymbol{y})|| \leq 2Cd. \tag{36}$$

Third, notice by directly applying Theorem 1 and replacing the Monte Carlo explanation by the high-quality explanation, we have that

$$\lambda_i^{\text{opt}} = \frac{\sum_{\substack{(\boldsymbol{x},\boldsymbol{y})\in\mathcal{D}_{\text{val}} \\ s_h(\boldsymbol{x})\in Q_i}} \langle \mathsf{MC}^n(\boldsymbol{x},\boldsymbol{y}) - \mathsf{Target}(\boldsymbol{x},\boldsymbol{y}), \mathsf{MC}^n(\boldsymbol{x},\boldsymbol{y}) - \mathsf{Amor}(\boldsymbol{x},\boldsymbol{y})\rangle}{\sum_{\substack{(\boldsymbol{x},\boldsymbol{y})\in\mathcal{D}_{\text{val}} \\ s_h(\boldsymbol{x})\in Q_i}} ||\mathsf{MC}^n(\boldsymbol{x},\boldsymbol{y}) - \mathsf{Amor}(\boldsymbol{x},\boldsymbol{y})||^2}. \tag{37}$$

Hence, we can write $\lambda_i^{\text{opt}} - \lambda_i$ as

$$|\lambda_i^{\text{opt}} - \lambda_i| \tag{38}$$

$$= \left|\frac{\sum_{\substack{(\boldsymbol{x},\boldsymbol{y})\in\mathcal{D}_{\text{val}} \\ s_h(\boldsymbol{x})\in Q_i}} \langle \mathsf{MC}^{n'}(\boldsymbol{x},\boldsymbol{y}) - \mathsf{Target}(\boldsymbol{x},\boldsymbol{y}), \mathsf{MC}^n(\boldsymbol{x},\boldsymbol{y}) - \mathsf{Amor}(\boldsymbol{x},\boldsymbol{y})\rangle}{\sum_{\substack{(\boldsymbol{x},\boldsymbol{y})\in\mathcal{D}_{\text{val}} \\ s_h(\boldsymbol{x})\in Q_i}} ||\mathsf{MC}^n(\boldsymbol{x},\boldsymbol{y}) - \mathsf{Amor}(\boldsymbol{x},\boldsymbol{y})||^2}\right| \tag{39}$$

$$\leq \frac{\left(\sum_{\substack{(\boldsymbol{x},\boldsymbol{y})\in\mathcal{D}_{\text{val}} \\ s_h(\boldsymbol{x})\in Q_i}} ||\mathsf{MC}^{n'}(\boldsymbol{x},\boldsymbol{y}) - \mathsf{Target}(\boldsymbol{x},\boldsymbol{y})||_2^2 ||\mathsf{MC}^n(\boldsymbol{x},\boldsymbol{y}) - \mathsf{Amor}(\boldsymbol{x},\boldsymbol{y})||_2^2\right)^{1/2}}{\sum_{\substack{(\boldsymbol{x},\boldsymbol{y})\in\mathcal{D}_{\text{val}} \\ s_h(\boldsymbol{x})\in Q_i}} ||\mathsf{MC}^n(\boldsymbol{x},\boldsymbol{y}) - \mathsf{Amor}(\boldsymbol{x},\boldsymbol{y})||^2}, \tag{40}$$

where the last inequality (40) comes from the Cauchy–Schwarz inequality. Denote the denominator in (40) by $\Delta$, i.e.,

$$\sum_{\substack{(\boldsymbol{x},\boldsymbol{y})\in\mathcal{D}_{\text{val}} \\ s_h(\boldsymbol{x})\in Q_i}} ||\mathsf{MC}^n(\boldsymbol{x},\boldsymbol{y}) - \mathsf{Amor}(\boldsymbol{x},\boldsymbol{y})||^2 = \Delta.$$

Lastly, notice that $\mathsf{MC}^{n'}(\boldsymbol{x},\boldsymbol{y})$ is sampled independently of $\mathsf{MC}^n(\boldsymbol{x},\boldsymbol{y})$ and that $\mathsf{Target}(\boldsymbol{x},\boldsymbol{y})$ is deterministic. Therefore:

$$\Pr[|\lambda_i^{\text{opt}} - \lambda_i| \geq \epsilon] \tag{41}$$

$$\leq \Pr\left[\frac{\left(\sum_{\substack{(\boldsymbol{x},\boldsymbol{y})\in\mathcal{D}_{\text{val}} \\ s_h(\boldsymbol{x})\in Q_i}} ||\mathsf{MC}^{n'}(\boldsymbol{x},\boldsymbol{y}) - \mathsf{Target}(\boldsymbol{x},\boldsymbol{y})||_2^2 ||\mathsf{MC}^n(\boldsymbol{x},\boldsymbol{y}) - \mathsf{Amor}(\boldsymbol{x},\boldsymbol{y})||_2^2\right)^{1/2}}{\sum_{\substack{(\boldsymbol{x},\boldsymbol{y})\in\mathcal{D}_{\text{val}} \\ s_h(\boldsymbol{x})\in Q_i}} ||\mathsf{MC}^n(\boldsymbol{x},\boldsymbol{y}) - \mathsf{Amor}(\boldsymbol{x},\boldsymbol{y})||^2} \geq \epsilon\right]$$

$$\tag{42}$$

$$\leq \Pr\left[\frac{\sum_{\substack{(\boldsymbol{x},\boldsymbol{y})\in\mathcal{D}_{\text{val}} \\ s_h(\boldsymbol{x})\in Q_i}} ||\mathsf{MC}^{n'}(\boldsymbol{x},\boldsymbol{y}) - \mathsf{Target}(\boldsymbol{x},\boldsymbol{y})||_2^2 ||\mathsf{MC}^n(\boldsymbol{x},\boldsymbol{y}) - \mathsf{Amor}(\boldsymbol{x},\boldsymbol{y})||_2^2}{\Delta^2} \geq \epsilon^2\right] \tag{43}$$

$$\leq \Pr\left[\frac{\sum_{\substack{(\boldsymbol{x},\boldsymbol{y})\in\mathcal{D}_{\text{val}} \\ s_h(\boldsymbol{x})\in Q_i}} ||\mathsf{MC}^{n'}(\boldsymbol{x},\boldsymbol{y}) - \mathsf{Target}(\boldsymbol{x},\boldsymbol{y})||_2^2||\mathsf{MC}^n(\boldsymbol{x},\boldsymbol{y}) - \mathsf{Amor}(\boldsymbol{x},\boldsymbol{y})||_2^2}{\Delta^2} \geq \epsilon^2 \,\middle|\, \Delta \leq \frac{\mu}{2}\right]$$

$$\times \Pr\left[\Delta \leq \frac{\mu}{2}\right]$$

$$+ \Pr\left[\frac{\sum_{\substack{(\boldsymbol{x},\boldsymbol{y})\in\mathcal{D}_{\text{val}} \\ s_h(\boldsymbol{x})\in Q_i}} ||\mathsf{MC}^{n'}(\boldsymbol{x},\boldsymbol{y}) - \mathsf{Target}(\boldsymbol{x},\boldsymbol{y})||_2^2||\mathsf{MC}^n(\boldsymbol{x},\boldsymbol{y}) - \mathsf{Amor}(\boldsymbol{x},\boldsymbol{y})||_2^2}{\Delta^2} \geq \epsilon^2 \,\middle|\, \Delta > \frac{\mu}{2}\right]$$

$$\times \Pr\left[\Delta > \frac{\mu}{2}\right] \tag{44}$$

$$\leq \Pr\left[\sum_{\substack{(\boldsymbol{x},\boldsymbol{y})\in\mathcal{D}_{\text{val}} \\ s_h(\boldsymbol{x})\in Q_i}} ||\mathsf{MC}^{n'}(\boldsymbol{x},\boldsymbol{y}) - \mathsf{Target}(\boldsymbol{x},\boldsymbol{y})||_2^2||\mathsf{MC}^n(\boldsymbol{x},\boldsymbol{y}) - \mathsf{Amor}(\boldsymbol{x},\boldsymbol{y})||_2^2 \geq \epsilon^2\frac{\mu^2}{4}\right]$$

$$+ \Pr\left[\Delta \leq \frac{\mu}{2}\right] \tag{45}$$

$$\leq e^{\frac{-\mu^2|Q_i|}{4Cd}} + e^{\frac{-\mu^4\epsilon^4|Q_i|}{400Cd}}. \tag{46}$$

Where the inequality in (42) is a direct application of 40, the inequality in (44) comes from simply conditioning, the inequality in (45) comes from the fact that probabilities are bounded by one getting rid of the first term in (45) (first out of lines) and the fourth term in (45) (forth out of lines) and the fact that $\mathsf{MC}^{n'}(\boldsymbol{x},\boldsymbol{y})$ is sampled independently of $\mathsf{MC}^n(\boldsymbol{x},\boldsymbol{y})$ and that $\mathsf{Target}(\boldsymbol{x},\boldsymbol{y})$ is deterministic. Finally, the last inequality in (46) comes from applying (30) and (35).

Hence, from (46), we conclude that

$$\Pr[|\lambda_i^{\texttt{opt}} - \lambda_i| \geq \epsilon] \leq e^{\frac{-\mu^2|Q_i|}{4Cd}} + e^{\frac{-\mu^4\epsilon^4|Q_i|}{400Cd}}. \tag{47}$$

$\square$

**Proposition 2** (Coverage for Inference Budget). *Let $N_{budget} \geq 1$ be the set inference budget, and assume that the Monte Carlo method $\mathsf{MC}^n(\boldsymbol{x},\boldsymbol{y})$ uses $n$ model inferences. Then, the coverage level $\alpha$ should be chosen such that*

$$\operatorname*{argmin}_{\alpha\in[0,1]}\left\{\mathbb{E}\left[N(\mathsf{SE}(\boldsymbol{x},\boldsymbol{y}))\right] \leq N_{budget}\right\} = \frac{n+1-N_{budget}}{n}. \tag{48}$$

*Recall that Shapley Value Sampling with parameter $m$ performs $1 + dm$ inferences ($\boldsymbol{x} \in \mathbb{R}^d$), and Kernel Shap with parameter $m$ performs $m$ inferences.*

*Proof.* Let $\alpha \in [0,1]$, then an $\alpha$ portion of examples receive explanations from the amortized explainer, i.e., they receive one inference, and $1 - \alpha$ portion of examples receive explanations with initial guess, i.e., $n$ model inferences. Therefore, the expected number of model inferences per instance is given by (49).

$$\mathbb{E}\left[N(\mathsf{SE}(\boldsymbol{x},\boldsymbol{y}))\right] = \alpha + (1-\alpha)(n+1) \tag{49}$$

In order for the inference budget to be followed, it is necessary that

$$\mathbb{E}\left[N(\mathsf{SE}(\boldsymbol{x},\boldsymbol{y}))\right] = \alpha + (1-\alpha)(n+1) \leq N_{\text{budget}}. \tag{50}$$

From (50), we conclude that:

$$\alpha \geq \frac{n+1-N_{\text{budget}}}{n}, \tag{51}$$

Hence,

$$\operatorname*{argmin} \alpha \in [0,1] \left\{\mathbb{E}\left[N(\mathsf{SE}(\boldsymbol{x},\boldsymbol{y}))\right] \leq N_{\text{budget}}\right\} = \frac{n+1-N_{\text{budget}}}{n}. \tag{52}$$

$\square$

