# OpenReview forum: "Selective Explanations"
_NeurIPS.cc/2024/Conference — NeurIPS 2024 poster_

### Official Review · Reviewer_xNyJ · 2024-06-25

**Soundness:** 2
**Presentation:** 2
**Contribution:** 2
**Rating:** 4
**Confidence:** 4

**Summary:**

The paper presents a novel framework for explaining black box models through a selective explainer. The selective explainer unifies two different explanation methods: the former relies on amortized explainers, which are easier to compute but provide lower-quality explanations; the latter can provide higher-quality explanations but is more costly to compute. By identifying when the first method is enough to have a good explanation, the selective explainer can trade off (depending on the user’s needs) when relying on amortized explainers or a more complex explanation method. Experiments on benchmark data show the effectiveness of the approach.

**Strengths:**

1. The paper considers an interesting problem, i.e. how to combine cheap-to-obtain explanations and expensive-to-obtain ones;
2. The theoretical analysis seems sound.

**Weaknesses:**

w1. The methodology is based on never-discussed assumptions, i.e. a more uncertain/less stable explanation is a low-quality explanation. I think the authors should discuss this assumption further. Moreover, I guess such a measure is potentially correlated with whether we explain correct or incorrect ML predictions, which is something that is never discussed/taken into account and lacks appropriate references.
w2.  The terminology is misleading: the paper calls expensive-to-obtain explanations high-quality explanations. I think this misleads the reader because there is never an actual comparison between expensive explanations and their actual quality in explaining the underlying ML model.
w3. The experimental evaluation is limited to quantitative measures. I think the paper would greatly benefit from including a user study, as humans must evaluate the quality of explanations provided by the novel selective explainer. For example, [Longo et al., 2024] provide a few reasons why human user studies should always be conducted when considering XAI methods.
w4. The paper does not take into account relevant related work on uncertainty quantification of explanation methods. For instance, see [Zhao et al, 2021] and [Slack et al., 2021].

[Longo et al., 2024 ] Longo, L., Brcic, M., Cabitza, F., Choi, J., Confalonieri, R., Del Ser, J., Guidotti, R., Hayashi, Y., Herrera, F., Holzinger, A. and Jiang, R., 2024. Explainable artificial intelligence (XAI) 2.0: A manifesto of open challenges and interdisciplinary research directions. Information Fusion, p.102301.
[Slack et al, 2021] Dylan Slack, Anna Hilgard, Sameer Singh, Himabindu Lakkaraju:
Reliable Post hoc Explanations: Modeling Uncertainty in Explainability. NeurIPS 2021: 9391-9404
[Zhao et al., 2021] Xingyu Zhao, Wei Huang, Xiaowei Huang, Valentin Robu, David Flynn:
BayLIME: Bayesian local interpretable model-agnostic explanations. UAI 2021: 887-896

**Questions:**

Q1 - What is a low-quality explanation under the proposed framework? I think this is the core question underlying the work. However, the authors are not directly addressing this question, as, in my opinion, assessing the low quality of a certain explanation requires defining who is the final user of the explanation.

Q2 -  As far as I know, an unstable explanation might be due to the underlying uncertainty around a prediction, but also due to the approximations required to compute the explanation. Is there a way to disentangle these two kinds of uncertainty? I think this is something that the method should take into consideration.

Q3—As discussed in W2, what the authors call a high-quality explanation seems to be obtained by a more costly (as it relies on some exact computation) explanation method. Is my understanding correct? If so, how do you ensure that the high quality explanation is a good explanation of why a certain prediction has been made?

A few other details:

Line 80: you refer to explanations with initial guesses.  I would personally prefer to have the definition here rather than in the next section.
Lines 152-154: can you clarify and discuss the drawbacks of this choice?

---

> ### Author Rebuttal · Authors · 2024-08-07
>
> Thank you for your review! It was really insightful and will definitely improve the final version of our paper. We will implement your suggested changes, and hope you engage with us during the discussion period. We address your comments below.
>
> **Q1) “The terminology is misleading: the paper calls expensive-to-obtain explanations high-quality explanations.”**
>
> This is a great point. We do agree with you that expensive-to-obtain explanations is a more precise term and will change “high-quality” to “expensive-to-obtain” everywhere in the revised version of our paper. Throughout the paper we assume that “high-quality” explanations are the ones computed until convergence – for example, SHAP with exponentially many inferences.
> These computationally expensive SHAP explanations have been extensively tested both quantitatively and via user studies [1, 2] – although SHAP does have limitations as well [3]. We will add the following paragraph on the quality of expensive-to-obtain explanations to the limitations section:
>
> “Selective explanations aim to approximate expensive-to-obtain explanation methods such as SHAP with an exponential number of inferences. Therefore, its efficacy is limited to the performance of the explanation method being approximated. The general methodology behind Selective Explanations is flexible, and can be applied to approximate other expensive-to-obtain feature attribution methods. We encourage the reader to use Selective Explanations to approximate the computationally expensive explanation method that best fits their application, noting that, ultimately, the quality of explanation will be limited by the chosen approximand.”
>
> **Q2) “The methodology is based on never-discussed assumptions, i.e. a more uncertain/less stable explanation is a low-quality explanation.”**
>
> Thank you for raising this important point. We address it in the global response to all reviewers due to its importance, and we kindly ask you to refer to the text above (see answer to Q2 above). TL;DR: Figure 3 illustrates the relationship between uncertainty and explanation quality, showing that explanations with the lowest uncertainty also have the lowest MSE (and vice-versa), where MSE is with respect to "higher-quality explanations" (SHAP computed until convergence). We agree that this correlation would be better supported by more direct evidence. To address this, we will add a new Table 1 (see attached rebuttal PDF) to the main paper, which presents Pearson and Spearman correlations between our uncertainty measures and MSE. This table provides evidence of strong positive correlations and monotonic relationships between MSE and uncertainty estimates, particularly for language datasets. The monotonic relationship underlies our approach's effectiveness in identifying lower-quality explanations by using uncertainty estimates. For more details, please refer to the global answer (Q2).
>
> **Q3) “The experimental evaluation is limited to quantitative measures.”**
>
> The main reason for the lack of user evaluation is that our main goal is to approximate widely-studied yet expensive-to-obtain explanation methods such as LIME (with a large number of model inferences) and SHAP (computed until convergence) [1, 2]. Given their established results, we follow the methodology in recent papers focused on accelerating attribution methods [5 – 8] and report both MSE and Spearman’s Correlation to these expensive-to-obtain explanations. When MSE is low and Spearman Correlation is high with converged SHAP explanations (which our Selective Explanation method achieves), the performance of our method is essentially the same as SHAP.
>
> **Q4) “The paper does not take into account relevant work on uncertainty quantification of explanation methods. For instance, see [Zhao et al, 2021] and [Slack et al., 2021].”**
>
> Thanks for suggesting these papers. We'll discuss them in our related work section. Our approach differs in two key ways from the mentioned papers:
>
> 1. These papers study uncertainty for SHAP and LIME variants. We calculate uncertainty for the amortized explainer - a model that takes data points as input and directly outputs explanations with only one inference.
>
> 2. Main contribution: Our focus isn't on the uncertainty measures themselves; they only serve the intermediate purpose of finding amortized explanations that severely differ from expensive-to-obtain explanations.
>
> **Q5)  “As far as I know, an unstable explanation might be due to the underlying uncertainty around a prediction, but also due to the approximations required to compute the explanation. Is there a way to disentangle these two kinds of uncertainty?”**
>
> We also believe that prediction uncertainty and approximations are sources of unstable explanations. However, we are computing the uncertainty for the amortized explainer, not the black-box model being explained ($h$ in our notation). Our method prioritizes practical improvement of the approximation of expensive-to-obtain explanations over the disentanglement of uncertainty sources – which we believe is an important direction of future work.
>
> [1] Lundberg et al. A Unified Approach to Interpreting Model Predictions. NIPS 2017.
>
> [2] Antwarg et al. Explaining Anomalies Detected by Autoencoders Using SHAP. Expert Syst. Appl. Vol 186.
>
> [3] Slack et al. Fooling LIME and SHAP: Adversarial Attacks on Post hoc Explanation Methods. AIES 2020.
>
> [4] Ribeiro et al. "Why Should I Trust You?": Explaining the Predictions of Any Classifier. KDD 2016.
>
> [5] Jethani et al. FastSHAP: Real-Time Shapley Value Estimation. ICLR 2022.
>
> [6] Schwab and Karlen. CXPlain: Causal Explanations for Model Interpretation under Uncertainty. NeurIPS 2019.
>
> [7] Yang et al. Efficient Shapley Values Estimation by Amortization for Text Classification. ACL 2023.
>
> [8] Covert et al. Stochastic Amortization: A Unified Approach to Accelerate Feature and Data Attribution. arXiv:2401.15866.

---

> > ### Comment · Reviewer_xNyJ · 2024-08-13
> >
> > Thank you to the authors for answering my questions. After reading all the reviews and rebuttals, I still have concerns and I do not change my opinion.

---

> > > ### Author Response · Authors · 2024-08-13
> > >
> > > Thank you for your response.
> > >
> > > Could you please share your remaining questions and concerns?
> > > We still believe we answered all your concerns in the rebuttal and would like to know why, in your opinion, they were not addressed.
> > >
> > > Thank you.

---

### Official Review · Reviewer_tCbB · 2024-07-09

**Soundness:** 3
**Presentation:** 3
**Contribution:** 3
**Rating:** 7
**Confidence:** 4

**Summary:**

This paper introduces a novel method called selective explanations for improving the efficiency and accuracy of feature attribution methods for black-box machine learning models. The key contributions are:

- A zero-cost proxy to evaluate the adversarial robustness of deep neural networks without training.
- A selective explanation method that detects when amortized explainers generate low-quality explanations and improves them using explanations with initial guess.
- An optimization approach for combining amortized and Monte Carlo explanations to improve explanation quality.
- Comprehensive evaluation on both tabular and language model datasets, demonstrating improved explanation quality with reduced computational cost.

The authors argue that their method addresses the challenges of computational expense in existing feature attribution methods while maintaining or improving explanation quality.

**Strengths:**

- Novel approach: The selective explanation method offers a new paradigm for balancing efficiency and accuracy in feature attribution.
- Theoretical foundation: The paper provides rigorous mathematical formulations and proofs for key components.
- Comprehensive evaluation: The method is tested on multiple datasets and model types, with comparisons to various baselines.
- Practical impact: The approach significantly reduces computational cost while maintaining or improving explanation quality.
- Flexibility: The method can be applied to different types of feature attribution techniques and model architectures.

**Weaknesses:**

- Limited exploration of very large models: While the method is tested on language models, it's not clear how well it scales to extremely large models (e.g., GPT-3 scale).
- Dependence on amortized explainers: The method's effectiveness relies on the quality of the underlying amortized explainer.
- Computational overhead: While more efficient than full Monte Carlo methods, the selective approach still requires additional computation compared to pure amortized methods.
- Sensitivity to hyperparameters: The impact of various hyperparameters (e.g., uncertainty thresholds, combination function parameters) is not thoroughly explored.

**Questions:**

- How does the performance of selective explanations scale with extremely large models (e.g., models with billions of parameters)?
- Have you explored using more advanced uncertainty estimation techniques, such as those based on Bayesian neural networks?
- How sensitive is the method to the choice of amortized explainer? How might it perform with different types of amortized explainers?
- Could the selective explanation approach be extended to other types of explanation methods beyond feature attribution (e.g., example-based explanations)?
- How might the method be adapted to handle streaming data or online learning scenarios where the underlying model is continuously updating?

**Limitations:**

The authors discuss limitations of their work at the end of Section 6. They acknowledge that the method is currently focused on Shapley values and has been primarily tested on specific types of models and datasets. They also note potential challenges in applying the method to image classifiers. These limitations are reasonably addressed, though a more detailed discussion of potential failure modes or edge cases could have been beneficial.

---

> ### Author Rebuttal · Authors · 2024-08-07
>
> Thank you for your review. We appreciate that you found our work novel. We also appreciate that you found our evaluation comprehensive and our method flexible. We answer your questions and comments next.
>
> **Q1)  “How does the performance of selective explanations scale with extremely large models (e.g., models with billions of parameters)?”**
>
> Our results indicate that Selective explanations scale well to very large models. We observed a better performance improvement when using Selective Explanation for larger models - the language models for text classification (Figures 3 and Figure 4 (c) and (d)). This is because we use the higher quality embeddings of these larger models to train the amortized explainer and the uncertainty metrics.
>
> **Q2) “Have you explored using more advanced uncertainty estimation techniques, such as those based on Bayesian neural networks?”**
>
> This is an excellent suggestion. We agree with you that other techniques for uncertainty quantification such as those based on Bayesian neural networks could potentially improve uncertainty estimation by providing more calibrated uncertainty estimates. In fact, our Deep uncertainty metric (Eq. 4) is directly inspired by this approach. However, Bayesian methods induce a higher cost that may be prohibitive in some applications. For example using ensembles of models as in deep ensembles [1] requires retraining the same model pipeline multiple times – a limitation we also observe in our proposed Deep uncertainty metric (Eq. 4). Moreover, our ultimate goal is to select the examples that receive higher (lower) quality explanations relative to a computationally-expensive-to-obtain method such as SHAP. In Learned uncertainty we “learn” from data which points will receive higher (lower) quality explanations instead of using a Bayesian approach. This leads to a method that is optimized to achieve our goal because it is trained to predict the examples with higher MSE relative to the high-quality explanations and decrease the computational overhead by only training a model once.
>
> [1] Lakshminarayanan et al. Simple and Scalable Predictive Uncertainty Estimation using Deep Ensembles. NIPS 2017.
>
>
> **Q3) “How sensitive is the method to the choice of amortized explainer? How might it perform with different types of amortized explainers?”**
>
> The main objective of selective explanations is to improve the quality of the explanations provided by both amortized and Monte Carlo explainers by combining both methods when the amortized explainer fails. Therefore, in the case that the amortized explainer has poor performance, selective explanations would require more usage of the Monte Carlo explainer to achieve favorable performance (i.e., small MSE from high-quality explanations), thus increasing the computational cost per explanation. Hence, the choice of amortized explainer heavily impacts the quality vs. computational cost of selective explanations.
>
> **Q4) “Could the selective explanation approach be extended to other types of explanation methods beyond feature attribution (e.g., example-based explanations)?”**
>
> The core idea of selective explanations is identifying low-quality predicted explanations and using a more computationally expensive method to improve their quality. Therefore, selective explanations could be extended to other explanation types. For example, with example-based explanations, an uncertainty metric could be developed to identify cases where selected examples are likely to be unrepresentative, and thus a different explanation method should be used.
>
> **Q5) “How might the method be adapted to handle streaming data or online learning scenarios where the underlying model is continuously updating?”**
>
> To handle streaming data or online learning, the components of the selective explanation method would need to be updated incrementally. Specifically, the amortized explainer, the combination function (Eq. 12), and the uncertainty measures (Eq. 4 and 5) would need to be continuously updated to reflect the changes in the model being explained.
>
> **Q6) “Computational overhead: While more efficient than full Monte Carlo methods, the selective approach still requires additional computation compared to pure amortized methods.”**
>
> The selective explanation method combines amortized and Monte Carlo explanations, introducing an additional computational cost. However, we believe this is a feature instead of a flaw, since it allows flexibility in choosing the fraction of samples that receives extra computations. Figure 5 shows that selective explanations significantly improve the performance of the amortized explainer, especially for the samples with worst-performing explanations. For example, Figure 5 (a) demonstrates that providing explanations with an initial guess to 50% of the data improves the Spearman’s correlation of the worst 10% amortized explanations from 0 (no correlation) to almost 0.6 (strong correlation).

---

> > ### Comment · Reviewer_tCbB · 2024-08-14
> > **Response to Author Rebuttal**
> >
> > I appreciate the detailed responses provided by the authors, and my concerns are well addressed. I would like to keep the original accept rating.

---

### Official Review · Reviewer_kYJk · 2024-07-10

**Soundness:** 3
**Presentation:** 3
**Contribution:** 3
**Rating:** 5
**Confidence:** 4

**Summary:**

This paper proposed a feature attribution method that detects when amortized explainers generate low-quality explanations and improves the explanations with their linear interpolation of themselves and expensive high-quality explanations.
To detect the low-quality explanations of the amortized explainers, the proposed method measures the uncertainty of the explanations.
Then, based on a selection function defined by the uncertainty, it selects either using the explanations by the amortized explainers or using the linear interpolation of the explanations by the amortized explainers and Monte Carlo explainers.
The experiments were conducted on tabular and text datasets, and the experimental results showed that the proposed method achieved better accuracy of explanations than using the amortized explainers and the Monte Carlo explainers solely.

**Strengths:**

- This work is the first to propose a feature attribution method in the "selective" setting, which can improve the quality of the explanations with Monte Carlo explanations when amortized explainers generate low-quality explanations. Since the proposed method is an extension of selective classification and regression, which are successful and well-studied approaches, it is easy to imagine it working well.
- The formalization of the proposed method is almost appropriate except that I am concerned about the learned uncertainty (5), and it is described clearly.

**Weaknesses:**

- In common, the quality of explanations has a trade-off between computational efficiency and accuracy. Although the experimental results show that the proposed method can improve the accuracy of explanations, those on computational efficiency are not investigated.
- The generated explanations are quantitatively evaluated. However, due to the lack of qualitative evaluation, it is unclear how good the explanations generated by the proposed method are actually from a user perspective.

The following are a minor point:
- If (5) is MSE, should use $\|| \cdot \||$ instead of $| \cdot |$.

**Questions:**

- **Learned uncertainty:** The loss $\ell(\mathrm{Amor(x;y)}, \mathrm{MC}^n(x,y))$ in (5) is the same as the objective of the amortized explainer $\mathrm{Amor(x;y)}$ in (3).
Therefore, if the amortized explainer is fitted enough to the training data, the loss $\ell$ is consistently near zero, resulting in the learned uncertainty function $s$ being consistently near zero, too.
This learned uncertainty function does not seem to work well at an inference phase. What is the justification for (5)?
- What does 'Random' in Figure 3 mean?

**Limitations:**

Yes.

---

> ### Author Rebuttal · Authors · 2024-08-07
>
> Thank you very much for your thoughtful review. Your feedback will positively impact the final version of our paper! We address your questions below.
>
> **Q1) ”Although the experimental results show that the proposed method can improve the accuracy of explanations, those on computational efficiency are not investigated.”**
>
> Thank you very much for this comment! Please kindly refer to the global answer to all reviewers above. TL;DR: the trade-off between computational complexity (given by number of model inferences) and quality of explanations (MSE from higher-quality explanations) is captured in Fig. 4 and 10 of the paper. Particularly, Fig. 10 (reproduced in the attached PDF) shows that our method performs very close to an oracle that "knows" which explainer to use for a given MSE and constraint on number of inferences. We will make this clearer by bringing Fig. 10 and the associated discussion to the main body of the paper instead of the appendix. Please, check the global answer (Q1) for more details.
>
> **Q2) “The generated explanations are quantitatively evaluated. However, due to the lack of qualitative evaluation, it is unclear how good the explanations generated by the proposed method are actually from a user perspective.”**
>
> Thank you for this great point. The main reason for the lack of user evaluation is that we are approximating expensive-to-obtain yet established explanation methods that were extensively tested and evaluated by humans such as LIME [1] and SHAP [2]. Given these established results, we follow the methodology in most papers that aim to approximate feature attribution methods [3, 4, 5, 6] and report both MSE and Spearman’s Correlation to SHAP explanations computed until convergence. For low MSE and high Spearman Correlation with SHAP (which our Selective Explanation method achieves), the performance of our method is essentially the same as SHAP. In other words, our goal is to approximate expensive-to-obtain SHAP explanations as closely as possible, while lowering the required number of inference and, thus, computational cost.
>
> Nevertheless, we acknowledge this point, and recognize that – since we aim to approximate methods such as SHAP – our method is ``high quality"' insofar that SHAP is high-quality, and will note this caveat in the paper.
>
> [1] Ribeiro et al. "Why Should I Trust You?": Explaining the Predictions of Any Classifier. KDD 2016.
>
> [2] Lundberg et al. A Unified Approach to Interpreting Model Predictions. NIPS 2017.
>
> [3] Jethani et al. FastSHAP: Real-Time Shapley Value Estimation. ICLR 2022.
>
> [4] Schwab and Karlen. CXPlain: Causal Explanations for Model Interpretation under Uncertainty. NeurIPS 2019.
>
> [5] Yang et al. Efficient Shapley Values Estimation by Amortization for Text Classification. ACL 2023.
>
> [6] Covert et al. Stochastic Amortization: A Unified Approach to Accelerate Feature and Data. Attribution. arXiv:2401.15866.
>
>
> **Q3) “If (5) is MSE, should use ||⋅||  instead of |⋅|.”**
>
> Thank you for this careful point. Notice that $\ell(\text{AMOR}(x;y), \text{MC}(x; y))$ is a scalar. Hence, $||.||$ and $|.|$ are equivalent.
>
> **Q4) “Learned uncertainty: The loss in (5) is the same as the objective of the amortized explainer in (3). Therefore, if the amortized explainer is fitted enough to the training data, the loss is consistently near zero, resulting in the learned uncertainty functions being consistently near zero, too. This learned uncertainty function does not seem to work well at an inference phase. What is the justification for (5)?”**
>
> Thank you for this great point. In the case there is overfitting, the selection function might not be sufficiently accurate (neither the amortized explainer), as you mentioned. In such cases, we advise the users to reserve a validation dataset to fit the uncertainty function. We highlight that we did not observe overfitting in any of our experiments where we train the uncertainty measure using the training dataset, and train the amortized explainers until convergence on the same training dataset, and evaluate their performance on a test set. However, this does not preclude the risk of overfitting in other settings. We will add your comment to our limitations section and advise on the use of a validation set instead of the same training dataset.
>
> **Q5) "What does 'Random' in Figure 3 mean?"**
>
> "Random" means that we select explanations uniformly at random instead of using the uncertainty metrics. This leads to an average MSE that is near the MSE of the amortized explainer and is  independent of the coverage. We will add this description in line 255 of our paper.

---

> > ### Comment · Reviewer_kYJk · 2024-08-14
> >
> > Thank you for your response. Some of my concerns have been allayed. I will maintain a positive score.

---

### Official Review · Reviewer_XcVA · 2024-07-12

**Soundness:** 3
**Presentation:** 3
**Contribution:** 3
**Rating:** 4
**Confidence:** 4

**Summary:**

The paper proposes a method, termed "Selective Explanations," aimed at improving the quality of explanations generated by amortized explainers in machine learning. The authors introduce a technique that detects low-quality explanations and employs a combination of amortized and Monte Carlo methods to enhance them. The approach leverages an "explanations with initial guess" technique, allowing a trade-off between computation speed and explanation quality. The proposed method is validated across different datasets, showing that it can improve the poorest quality explanations typically provided by amortized explainers.

**Strengths:**

1. The concept of using a selective approach to manage computational resources while improving explanation quality is compelling and timely.
2. The proposed method is intuitive and easy to understand.
3. The paper is well written.

**Weaknesses:**

1. The experimental design does not strongly support the major claims on the reduction of computational cost. Evaluations on the tradeoff between explanation quality and computational cost would be helpful.
2. The relationship between uncertainty and explanation quality is unclear, it would be better to have empirical or theoretical proof on their correlations. In addition, the proposed uncertainty measurement also introduces computational overhead when “run the training pipeline for the amortized explainer described in (3) k times” (line 139).

**Questions:**

See weaknesses.

**Limitations:**

Given the current state of the submission, the reasons to reject slightly outweigh the reasons to accept.

---

> ### Author Rebuttal · Authors · 2024-08-07
>
> Thank you for your thoughtful review. Your feedback will positively impact the updated version of our paper and we will include all of your comments. We also appreciate that you found our manuscript well-written, our method intuitive and easy to understand, and that our idea of Selective Explanations is compelling and timely. We address your comments below. We hope you can engage with us during the discussion period.
>
> **Q1)  “The experimental design does not strongly support the major claims on the reduction of computational cost. Evaluations on the tradeoff between explanation quality and computational cost would be helpful.”**
>
> We appreciate this comment and we kindly ask that you refer to Q1 in the global answer to all reviewers above. The TL;DR is: the trade-off between computational complexity (given by number of model inferences) and quality of explanations (MSE from high-quality explanations) is captured in Fig. 4 and 10 of the paper. In particular, Fig. 10 shows that our method significantly reduces the number of inferences required to achieve a given explanation quality in comparison with Monte Carlo explanations – specifically SVS. Remarkably, Fig. 10 shows that Selective Explanations performs very close to an oracle that "knows" which explainer to use for a given MSE and constraint on number of inferences. We will make this clearer by bringing Fig. 10 and the associated discussion to the main body of the paper instead of the appendix. Please, check the global answer (Q1) for more details.
>
> **Q2) “The relationship between uncertainty and explanation quality is unclear, it would be better to have empirical or theoretical proof on their correlations.”**
>
> Again, thank you for this thoughtful comment. Here, again, we kindly ask for you to refer to the global answer for all reviewers above and the attached pdf, which empirically shows a high correlation between uncertainty and explanation quality. TL;DR: Figure 3 shows the relationship between uncertainty and explanation quality by demonstrating that the amortized explanations with smallest uncertainty also have the smallest MSE from high-quality explanations and vice-and-versa. However, we agree with you that more direct evidence is needed. To address this, we computed this correlation, in a new Table 1 (attached in the PDF) and will add it to the main paper. This table presents positive Pearson and Spearman’s correlations between our uncertainty measures and MSE and a strong monotonic relationship between both quantities, especially for the language datasets. This evidence reinforces the effectiveness of our approach in detecting lower-quality explanations in the main tasks of interest. Again, please, check the global answer (Q2) for more details.
>
> **Q3)  “In addition, the proposed uncertainty measurement also introduces computational overhead when “run the training pipeline for the amortized explainer described in (3) k times” (line 139).”**
>
> We agree with you, the Deep Uncertainty measure introduces a computational overhead. Please note that this computational overhead is exactly the reason we proposed a second method, referred to as "Learned Uncertainty," which only needs to be trained once. Our experiments suggest that the Learned Uncertainty provides a better performance for selective explanations of language models – our main use case of interest. We will make this distinction clearer in the paper by adding the following paragraph after line 148:
>
> “To address the computational overhead of Deep Uncertainty and directly target low-quality explanations, we introduce Learned Uncertainty. This alternative method requires only one training run, drastically reducing computational costs. Learned Uncertainty is optimized to predict discrepancies between amortized and high-quality explanations.”
>
> We proposed Deep Uncertainty because this method is directly inspired by Deep Ensembles [1] used for distribution-free uncertainty quantification. It achieves favorable performance (see Figs. 3, 4, 5, and 6), but it indeed comes with a computational overhead. We introduce it because of its connection with Deep Ensembles traditionally used for uncertainty quantification.
>
> [1] Lakshminarayanan et al. Simple and Scalable Predictive Uncertainty Estimation using Deep Ensembles. NIPS 2017.

---

> > ### Comment · Reviewer_XcVA · 2024-08-13
> >
> > Thank you to the authors for answering my questions. After reading all the reviews and rebuttals, I still have concerns. The underlying assumption of this paper that "expensive-to-obtain" or computationally demanding explanations are "high-quality" lacks both empirical and theoretical validation. More discussion on this underlying assumption is critical. Furthermore, the definition of a "high-quality" explanation is not clearly defined. For instance, such explanations could be interpreted as more faithful to the model's decision-making process, more aligned with human preference, or more aligned with ground truth explanations.

---

> > > ### Author Response · Authors · 2024-08-13
> > >
> > > As we wrote to Reviewer xNyJ, we do agree that "expensive-to-obtain" explanations is a more precise term and will change "high-quality" to "expensive-to-obtain" everywhere in the revised paper.
> > >
> > > In our experiments, we use SHAP [1] with exponentially many computations as the "expensive-to-obtain" explanation. SHAP with exponentially many computations was already validated by previous literature (i) theoretically, (ii) empirically, and (iii) by user studies in the paper that proposed such explanations [1] and also in follow-up work [6, 7, 8] – its limitations have also been studied [9]. Although these explanations have many desired properties, they are computationally expensive. For this reason, a new stream of work that tries to approximate these explanations emerged [2, 3, 4, 5]. Please note that these papers do not argue on the quality of SHAP, but they do argue on how close their approximation is to converged explanations – as we also do.
> > >
> > > We hope to have addressed your concerns and are available to answer any further questions you may have.
> > >
> > >
> > > [1] Lundberg et al. A Unified Approach to Interpreting Model Predictions. NIPS 2017.
> > >
> > > [2] Jethani et al. FastSHAP: Real-Time Shapley Value Estimation. ICLR 2022.
> > >
> > > [3] Yang et al. Efficient Shapley Values Estimation by Amortization for Text Classification. ACL 2023.
> > >
> > > [4] Covert et al. Stochastic Amortization: A Unified Approach to Accelerate Feature and Data
> > > Attribution. arXiv:2401.15866.
> > >
> > > [5] Covert et al. Improving KernelSHAP: Practical Shapley Value Estimation via Linear Regression. PMLR 2021.
> > >
> > > [6] Yingchao. Explainable AI methods for credit card fraud detection: Evaluation of LIME and SHAP through a User Study. Dissertation 2021.
> > >
> > > [7] Salih et al. A Perspective on Explainable Artificial Intelligence Methods: SHAP and LIME. Advanced Intelligent Systems 2024.
> > >
> > > [8] Antwarg et al. Explaining Anomalies Detected by Autoencoders Using SHAP. Expert Syst. Appl. Vol 186.
> > >
> > > [9] Slack et al. Fooling LIME and SHAP: Adversarial Attacks on Post hoc Explanation Methods. AIES 2020.

---

### Author Rebuttal · Authors · 2024-08-07

### Global Rebuttal

Thank you very much to the reviewers for their effort! We are pleased that you found the paper well-written, the problem setting interesting, and our theoretical analysis sound (all reviewers), recognized that we are the first to propose "selective" feature attribution (reviewers kYjk and tCbB), and that our method is intuitive and easy to understand (reviewer XcVA).

Next, we answer two questions that were common across reviewers regarding computational cost and the relationship between MSE and uncertainty metrics.

**Q1) What is the tradeoff between explanation quality and computational cost?**

We thank reviewers XcVA and kYJk for bringing this up. The trade-off between computational cost (given by number of inferences) and quality of explanations (MSE w.r.t. a converged SHAP explanation) is captured in Fig. 4 and 10 of the paper. In particular, Fig. 10 (replicated in the rebuttal PDF) shows that the Selective Explanation method's computational cost is close to an oracle that "knows" which explainer to use for a given MSE. We will bring Fig. 10 and the associated discussion into the main body of the paper. We give details next.

1. Fig. 10 compares Selective Explanations with Monte Carlo methods in terms of computational cost (x-axis: number of model inferences) and explanation quality (y-axis: MSE w.r.t. high-quality explanations). The number of inferences serves as a proxy for computational cost, as GPU/CPU usage time scales linearly with it. Our results demonstrate that Selective Explanations achieve lower MSE than Monte Carlo methods for the same computational cost.

In our Selective Explanation method, we predict which samples should be routed to a more computationally expensive explanation method. We compare this method against an "oracle" who knows a priori how to optimally route samples in terms of MSE. We simulate this oracle by pre-computing SVS explanations with parameters 12, 25, and 50, and selecting the one with the smallest MSE from the target SHAP explanation. The oracle is simulated for comparison purposes only. Remarkably, Fig. 10 shows that selective explanations closely approximate the Oracle curve, indicating that, on these benchmarks, our method has a near-oracle trade-off between the number of inferences and MSE.

2. Fig. 4 also compares computational cost (percentile with recourse ($1 - \alpha$) in x-axis) and explanation quality (MSE from converged SHAP explanation, y-axis). Here, percentile with recourse is a proxy for computational cost because it controls the fraction of the dataset that will receive more computationally-expensive explanations, i.e., Monte Carlo or explanations with initial guess. These explanation methods are more costly than amortized methods because they require a larger number of inferences while the amortized explainer only requires one inference.

Fig. 4 demonstrates that Selective Explanations achieve lower MSE than Monte Carlo methods using the same number of inferences. In Fig. 4 (a) and (c), Selective Explanations can match the performance of the more expensive method while using only 50\% of the computational resources. Fig. 4 (b) and (d) show an even stronger result: Selective Explanations improve upon amortized explanations even when the more computationally expensive method has a higher MSE than the amortized method. This improvement is particularly significant for the lower-quality amortized explanations, as illustrated in Fig. 5.

**Q2) What is the relationship between our uncertainty metrics and explanation quality?**

We thank reviewers XcVA and xNyJ for this question. Figure 3 shows the relationship between our proposed uncertainty metrics (deep and learned uncertainty) and explanation quality (measured by MSE). MSE serves as a proxy for explanation quality as it measures the difference from SHAP explanations computed until convergence. Additionally, we included a new Table 1 (see attached PDF) showing Pearson’s (linear) and Spearman’s (monotonic) correlation measures between the uncertainty metrics and MSE, which will be added to the paper. These results show that our uncertainty metrics are consistently positively correlated with explanation quality across different tasks.

1. In Fig. 3, the x-axis represents coverage ($\alpha$), the fraction of examples with the lowest predicted uncertainty, while the y-axis shows the average MSE for these examples. For instance, with $\alpha = 25$\%, the y-axis shows the average MSE for the 25\% of examples with the lowest predicted uncertainty.

Fig. 3 indicates that examples with higher uncertainty metrics also have higher MSE because as coverage increases, the MSE of the selected explanations also increases. We compare our method to an "Oracle" that has access to the MSE of the amortized explanations and, when the coverage decreases, removes exactly those explanations with the highest MSE. Our approach closely matches the Oracle for both language tasks, Yelp Review and Toxigen datasets, demonstrating that our uncertainty measures are nearly optimal in these cases.

2. Table 1 (in the attached PDF) shows both Pearson and Spearman’s correlation coefficients between our uncertainty measures (Deep and Learned) and the explanation quality (measured by MSE) across the datasets. The table shows that our uncertainty measures achieve positive correlations across all datasets and uncertainty methods, indicating that as uncertainty increases, so does the MSE (lower explanation quality). We also observe strong correlations in many cases, particularly for the Learned uncertainty method on the Toxigen dataset (Pearson: 0.89, Spearman: 0.93). Moreover, Spearman’s correlation is constantly higher than Pearson’s, suggesting a strong monotonic relationship between uncertainty and explanation quality – which is our main interest since we aim to detect and rank lower-quality explanations.

---

### Decision · Program_Chairs · 2024-09-25

**Decision:**

Accept (poster)

**Comment:**

The paper presents a novel framework for a more flexible way of feature attribution in black box models, called the "selective explanations". The selective explainer tries to bridge two different explanation methods -- a) amortized explainers that are easier to compute but of lower quality, and b) the traditional higher-quality explanations that are costly to compute. The new method works simply by identifying when a) generates low-quality explanations, and improving them using an "explanations with initial guess" technique. In the end, the method allows a trade-off between computation speed and explanation quality.

The reviewers shared a similar set of questions/concerns: 1) explanation and computation tradeoff is missing; 2) unvalidated so-called "high-quality" explanations (which in the paper are interchangeably with "expensive" explanations; 3) the relationship between our uncertainty metrics and explanation quality is not discussed. For 1) there was a figure in the appendix that the authors plan to move to main text. For 2) the authors will stop suing the two terms interchangeably. But in my opinion the well established expensive methods such as SHAP is generally acknowledged to be high quality. A clarification in the text that this is what they regard as high-quality would be great. For 3), the authors added further experiments, which to me successfully addressed the concern.

To me, the initial draft was missing some important discussions, and thanks to the reviewers, the revised version is much better. However, two of the reviewers rating negatively seem to have an anchoring bias and refuse to update their scores, or engage further. To my eyes, the updated draft (with the promised changes) does meet the standard of NeurIPS acceptance.